# Distinct subtypes of proprioceptive dorsal root ganglion neurons regulate adaptive proprioception in mice

Haohao Wu [1], Charles Petitpré[1,8], Paula Fontanet [1,8], Anil Sharma [1], Carmelo Bellardita[1,2], Rolen M. Quadros[3], Paulo R. Jannig[4], Yiqiao Wang[1], J. Alexander Heimel [5], Kylie K. Y. Cheung[1], Simone Wanderoy[1], Yang Xuan[1], Konstantinos Meletis [1], Jorge Ruas [4], Channabasavaiah B. Gurumurthy [3,6], Ole Kiehn[1,2], Saida Hadjab [1] & François Lallemend [1,7 ✉]

Proprioceptive neurons (PNs) are essential for the proper execution of all our movements by providing muscle sensory feedback to the central motor network. Here, using deep single cell RNAseq of adult PNs coupled with virus and genetic tracings, we molecularly identify three main types of PNs (Ia, Ib and II) and find that they segregate into eight distinct subgroups. Our data unveil a highly sophisticated organization of PNs into discrete sensory input channels with distinct spatial distribution, innervation patterns and molecular profiles. Altogether, these features contribute to finely regulate proprioception during complex motor behavior. Moreover, while Ib- and II-PN subtypes are specified around birth, Ia-PN subtypes diversify later in life along with increased motor activity. We also show Ia-PNs plasticity following exercise training, suggesting Ia-PNs are important players in adaptive proprioceptive function in adult mice.

[1] Department of Neuroscience, Karolinska Institutet, Stockholm, Sweden. [2] Department of Neuroscience, University of Copenhagen, Copenhagen, Denmark. [3] Mouse Genome Engineering Core Facility, Department of Pharmacology and Experimental Neuroscience, College of Medicine, University of Nebraska Medical Center, Omaha, NE, USA. [4] Department of Physiology and Pharmacology, Karolinska Institutet, Stockholm, Sweden. [5] Cortical Structure & Function Group, Netherlands Institute for Neuroscience, Amsterdam, The Netherlands. [6] Department of Pharmacology and Experimental Neuroscience, College of Medicine, University of Nebraska Medical Center, Omaha, NE, USA. [7] Ming Wai Lau Centre for Reparative Medicine, Stockholm Node, Karolinska Institutet, Stockholm, Sweden. [8] These authors contributed equally: Charles Petitpré, Paula Fontanet. ✉email: francois.lallemend@ki.se

Proprioceptive feedback constantly monitors consequences of motor action. It is essential for the efficiency, precision, and robustness of movements in all motile animals and necessary for locomotor recovery in pathological conditions[1,2]. In mammals, the gatekeepers of the proprioceptive system are the proprioceptive neurons (PNs), residing in the dorsal root ganglia (DRG) along the spinal cord, projecting peripherally to skeletal muscles and continuously reporting to the central nervous system[3]. Three functional types of PNs have been identified from anatomical and physiological studies: the stretch-sensitive Ia- and II-PNs, which innervate muscle spindles (MSs), and the force-sensitive Ib-PNs, which innervate Golgi tendon organs (GTOs) (Fig. 1a)[3–5]. While this classic view on the hard-wired proprioceptive system has not been challenged for decades, the understanding of the central motor circuits has evolved[6,7], revealing its far more sophisticated organization to endow the rich repertoire of behaviors and high-level of plasticity to accommodate the ever-changing activities in mammals. It is thus unclear whether a larger diversity and plasticity of cell type and circuit is a basis for governing proprioceptive functions for accurate motor control.

Recent efforts in transcriptomic profiling of DRG neurons have generated a molecular catalog of somatosensory neurons and start to shed light on the transcriptional heterogeneity within some sensory types, e.g., nociceptors and mechanoreceptors[8–11]. Results from these studies demonstrate the existence of a dozen subtypes of nociceptive neurons, three subtypes of skin mechanoreceptors, and one homogenous population of PNs. Therefore, the understanding of the molecular diversity within PNs remains limited. This likely reflects the natural challenge of

molecular profiling of PNs, i.e., its low abundance among DRG neurons and the transcriptomic similarity between the subtypes. Here, combining deep scRNAseq of large number of PNs, mouse genetics, anatomical tracing, and exercise training, we report a highly specialized organization of the proprioceptive system comprising eight subtypes with distinct molecular profiles, spatial distribution and patterns of connectivity, revealing high levels of proprioceptive sensory inputs specialization. Moreover, their versatile composition in adult after sustained exercise training suggests plasticity with impact for sensory performance adaptability.

## Results

**Deep single-cell RNAseq reveals molecular diversity of adult PNs.** To obtain an enriched population of PNs amongst DRG neurons, we FAC sorted cells expressing tdTomato (TOM) from C5 to T1 DRG of postnatal stage 54 (P54) $PV^{Cre}$;Ai14 mice and processed them with Smart-seq2 protocol (Fig. 1b, c and Supplementary Fig. 1a)[12]. A total of 1109 PNs ($Whrn^+$, $Pvalb^+$, $Runx3^+$, $Ntrk3^+$ and $Etv1^+$) passed quality control with a high gene coverage (~11,000 genes detected per cell) (Supplementary Fig. 1b, c), allowing deep analysis of their molecular profiles. A small number of mechanoreceptors ($Ntrk2^+$ or $Ret^+$, $Pvalb^{low}$ and $Whrn^-$) were also identified and thus excluded from the subsequent analysis (Supplementary Fig. 1d). Transcriptomic analysis[13] (see "Methods" for details) revealed highly diversified picture of adult PNs, comprising eight molecularly distinct clusters (Fig. 1d), each with numerous genetic markers (Fig. 1e and

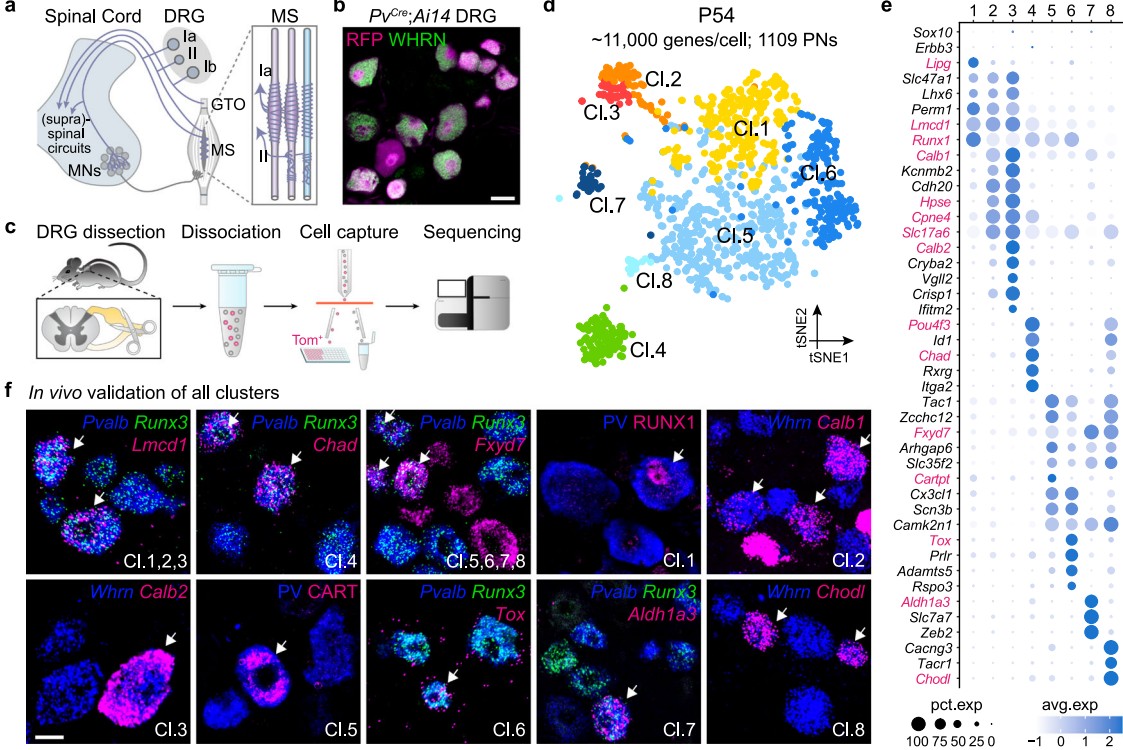

**Fig. 1 Deep scRNAseq reveals large heterogeneity among adult PNs. a** Schematic depiction of the central and peripheral innervation patterns of the three major PN types in the adult. **b** Brachial DRG section of P40 $PV^{Cre}$;Ai14 mouse stained for PN marker WHRN. Scale bar: 20 μm. **c** Schematic illustration of the workflow for scRNAseq of P54 PNs. **d** Adult PNs single-cell transcriptomes visualized with t-distributed stochastic neighbor embedding (tSNE), color-coded for the eight molecularly defined clusters. **e** Examples of marker genes expression in each cluster. The size of the circle reflects the proportion of the cells expressing the marker gene in a cluster, and the color intensity reflects its average expression level within that cluster. The markers used in subsequent analysis are indicated in magenta. **f** In vivo validation of the clusters of PNs by RNAscope and immunohistochemistry using the identified markers (magenta) on P54 brachial DRG sections. PNs are labeled by PV for immunohistochemistry, and $Pvalb$/$Runx3$ or $Whrn$ for RNAscope. Scale bar: 20 μm.

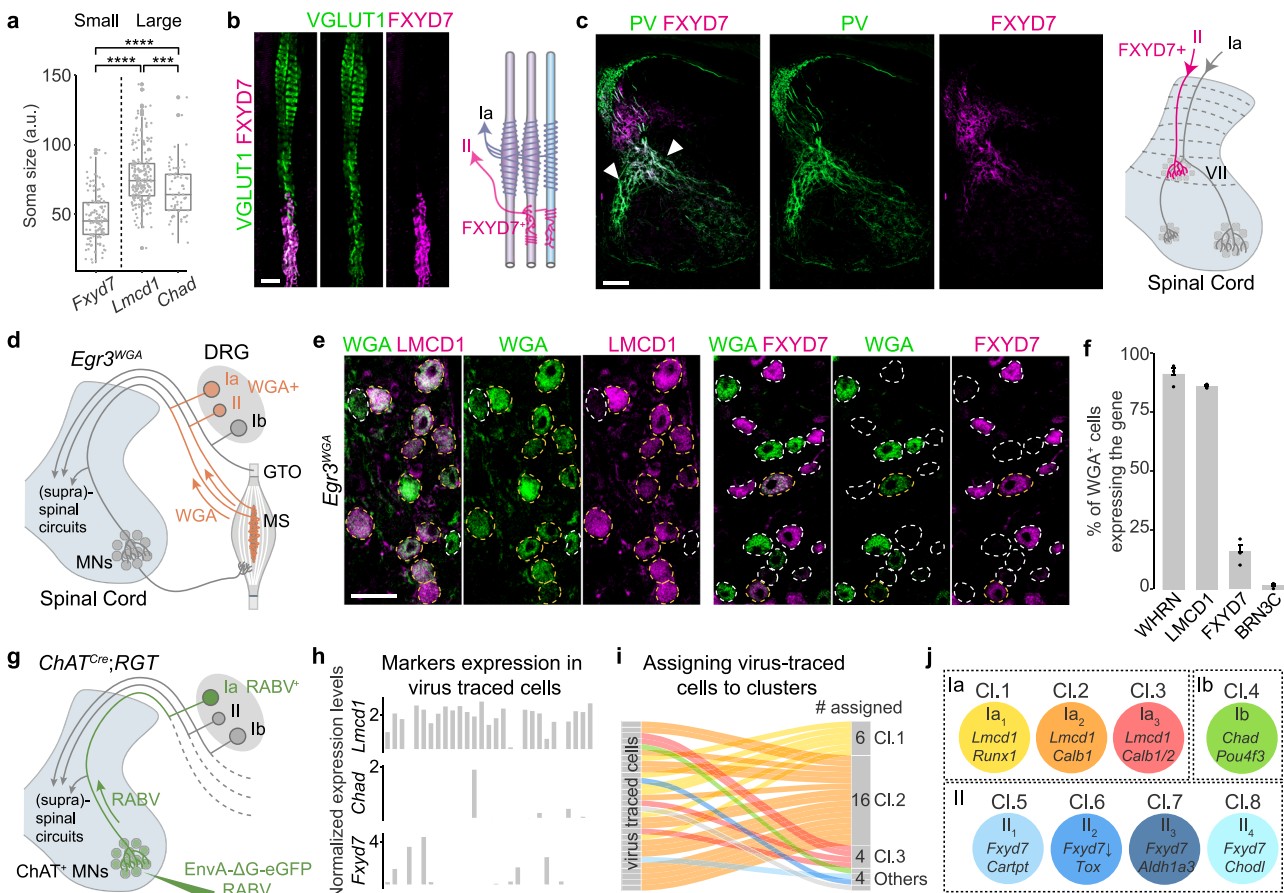

**Fig. 2 Identification of the major PN types. a** Quantification of cross-nucleus soma sizes of the three major PN populations depicted in box and whisker plot. Lower and upper hinges: first and third quartiles; the horizontal line: median; the whiskers extend to the value no further than 1.5 * IQR from the hinge; large dots: outliers. Each dot represents a cell: $Fxyd7^+$ PNs ($n = 124$ cells), $Lmcd1^+$ PNs ($n = 227$ cells), $Chad^+$ PNs ($n = 69$ cells). Two-tailed $t$-test, ****$p <$ 0.0001, ***$p < 0.001$. **b** Longitudinal section of muscle spindle (MS) stained for VGLUT1 and FXYD7. Scale bar: 20 µm. Right panel: schematic depiction of the finding. **c** Transverse section of P4 spinal cord stained for PV and FXYD7. Scale bar: 100 µm. Right panel: schematic depiction of the finding. **d** Genetic strategy to label MS-innervating PNs in $Egr3^{WGA}$ mice. **e** Representative image showing co-labeling of DRG neurons by WGA and LMCD1 or FXYD7 in DRG sections from P14 $Egr3^{WGA}$ mice. Scale bar: 50 µm. **f** Quantification of WGA$^+$ cells expressing WHRN, LMCD1, FXYD7, and BRN3C, respectively, in P14 brachial DRG ($n = 3$ animals). Data are presented as mean ± SEM; dots represent values from individual animals. **g** Trans-monosynaptic rabies virus tracing strategy to label Ia-PNs in adult $ChAT^{Cre};RGT$ mice. **h** Bar plots showing that the majority of the rabies virus infected PNs express $Lmcd1$ but not $Fxyd7$ or $Chad$. **i** Assignment of virus infected PNs to the identified eight PN clusters based on their overall transcriptomic similarity. **j** A summary of correspondence between the known functional types of PNs and the cell clusters identified by scRNAseq. Source data are provided as a Source Data file.

Supplementary Fig. 1e). Among those, we observed transcriptomic similarities within clusters (cl.) 1–3 ($Lmcd1^+$) and within cl.5–8 ($Fxyd7^+$), which together with the cl.4 ($Chad^+$) form three major groups (see "Methods" for details). We further validated that all eight clusters and three major groups exist in vivo using the newly identified markers (Fig. 1f).

**Molecular identification of Ia-, Ib-, and II-PNs.** To identify the main functional types of PNs (Ia, Ib, and II) among the molecularly distinct clusters, we first analyzed the three major groups of PNs as defined by the expression of the new marker genes $Lmcd1$ (cl.1-3), $Chad$ (cl.4), and $Fxyd7$ (cl.5-8), which together cover all PNs at brachial level. Combinatorial expression analysis of the three markers showed minimal overlapping between $Fxyd7$ and $Lmcd1$ or $Fxyd7$ and $Chad$, and few $Chad^+$ PNs expressing $Lmcd1$ but systematically at very low levels (Supplementary Fig. 2a, b). Morphologically, PN types can be distinguished by their axon diameter (medium: II-PNs: large: Ia- and Ib-PNs), which is proportional to their soma size and correlates with their conduction velocities[14,15]. We found that $Fxyd7^+$ PNs exhibited

on average smaller soma size compared to $Lmcd1^+$ and $Chad^+$ PNs (Fig. 2a and Supplementary Fig. 2c). In addition, FXYD7$^+$ PNs peripherally innervated MSs with endings invariably exhibiting "flower-spray"-like structure and positioning away from the central part of the MSs (Fig. 2c)[3,16] and centrally terminated at lamina VII of the spinal cord (Fig. 2c)[17,18]. These distinct attributes strongly suggest that FXYD7$^+$ PNs represent II-PNs.

To assign the two populations of large size PNs marked by the expression of $Lmcd1$ or $Chad$ to either Ia-PNs (innervating MSs) or Ib-PNs (innervating GTOs), we used the $Egr3^{WGA}$ mouse line in which wheat germ agglutinin (WGA) is expressed by the $Egr3^+$ intrafusal muscle fibers of MSs and back-traces Ia- and II- but not Ib-PNs in DRG (Fig. 2d)[19]. In this mouse line, the selective expression of WGA in the central part of MSs would preferentially trace afferents of Ia-PNs (Supplementary Fig. 2d). We found that 91% of WGA$^+$ cells were PNs, out of which the majority were LMCD1$^+$ (Fig. 2e, f) and 83% of LMCD1$^+$ PNs were WGA$^+$. In contrast, WGA was virtually absent from BRN3C$^+$ PNs (BRN3C, coded by $Pou4f3$, marks the $Chad^+$ cl.4, while we could not obtain a working CHAD antibody) (Fig. 2f), indicating that they represent Ib-PNs. We also observed 15% of

WGA[+] cells expressing FXYD7 (Fig. 2e, f), further supporting the type II identity of FXYD7[+] PNs. These results, together with the larger soma size of *Lmcd1*[+] PNs (Fig. 2a), strongly suggest that LMCD1[+] PNs are Ia-PNs. To confirm this, we set to obtain Ia-PNs transcriptome specifically using a rabies virus-mediated trans-synaptic tracing strategy to selectively label Ia-PNs following motor neurons infection in *ChAT*[Cre];*RGT* mice (Fig. 2g)[20]. Five days post infection, we performed scRNAseq on the isolated eGFP[+] neurons from the virus-infected DRG, showing that most of the virus-infected PNs (most, if not all, are Ia-PNs) expressed *Lmcd1*, but not *Fxyd7* or *Chad* (Fig. 2h) and 26 out of 30 virus-infected PNs were unbiasedly assigned to cl.1–3 using machine learning algorithm (Fig. 2i)[21], further confirming the type Ia identity of the LMCD1[+] PNs.

Altogether, we have identified the unique genetic signatures of the three major types of PNs (Ia, Ib, and II) and provided a molecular framework for PNs classification (Fig. 2j and Supplementary Dataset 1). Our scRNAseq analysis also reveals a large diversity of Ia- and II-PNs, implying a sophisticated organization of proprioceptive feedback from MSs.

### Functional subdivisions of PNs

*Main PN types*. The discovery of marker genes for the main types of PNs enables us to answer how PN types and their associated sensory feedback are distributed to different skeletal muscles that have distinct functions and performance requirements. We found that the proportion of the Ia-PNs and II-PNs was relatively constant across all spinal segments analyzed (Fig. 3a, b and Supplementary Fig. 3a). However, at the level of individual muscles, the composition of Ia- and II-PNs nerve endings within MSs varied dramatically: most MSs in biceps were either intermediate or complex (with one or two II-PN endings in addition to one Ia-PN ending), whereas MSs in triceps were mostly simple, i.e., devoid of II-PNs innervation (Supplementary Fig. 3b). This suggests that the type of MS feedback (velocity vs. length of stretch) transmitted to the central motor circuits varies between muscles.

While Ia- and II-PNs were found at all levels of the rostro-caudal axis, Ib-PNs, which innervate GTOs, were almost exclusive to the brachial and lumbar DRG that supply limb muscles in addition to axial muscles (Fig. 3b). In support of this, we did not observe any GTOs in the mouse intercostal and erector spinae muscles (thoracic axial muscles) using VGLUT1 staining. This suggests that axial muscles, in large part, lack GTOs and associated sensory feedback.

Altogether, these data indicate that Ia, Ib and II sensory afferents are allocated differently to muscles, suggesting distinct sensory feedback demands from the central nervous system for distinct muscles.

*PN subtypes*. We next set to understand the subdivisions within the main types of PNs through morphological and anatomical characterization, in order to address the functional relevance of the newly identified subtypes. Among Ia-PN subtypes, we observed that Ia$_1$-PNs exhibited relatively smaller soma sizes and higher abundance in all DRG analyzed (Fig. 3c, d), while Ia$_2$- and Ia$_3$-PNs exhibited larger soma sizes and could only be found in the DRG of brachial and lumbar segments innervating the limbs (Fig. 3c, d and Supplementary Fig. 3c, d). In line with this, using *Calb1*[dgCre];*Ai14* mice, the spinal VGLUT1[+]/RFP[+] terminal boutons of Ia$_2$- and Ia$_3$-PNs were systematically found in the lateral motor column (LMC) area (Fig. 3e), which is occupied by limb-innervating MNs. The analysis of gene expression profile of Ia$_2$- and Ia$_3$-PNs highlighted the genes sets related to synaptic proteins, ion channels, metabolic process and neurofilament

(Supplementary Fig. 3e), which together with their morphological and anatomical profile, suggest that they represent specific limb-innervating Ia-PN subtypes with higher activity demand and faster conduction velocity. Interestingly, Ia$_3$-PNs were specifically enriched in C6 at brachial level (Fig. 3d), suggesting that they innervate selective muscle targets. In support of this, using *Calb2*[Cre];*Ai14*, we found that in the forelimb, nerve endings of Ia$_3$-PNs were much more abundant in MSs of the dorsal distal muscles of the limb, which are principal extensors of the wrist and digits (Fig. 3f, g). More specifically, ~80% of MSs in ECR were innervated by Ia$_3$-PNs, while MSs in triceps were completely devoid of Ia$_3$-PNs innervation (Fig. 3g). Moreover, while Ia-PNs are known to innervate both nuclear bag and chain fibers, we observed that Ia$_3$-PNs specifically spiraled around the nuclear bag fibers (Fig. 3h), which are more sensitive to the dynamic changes of muscles. In comparison with Ia$_2$-PNs, Ia$_3$-PNs showed a particularly high expression of genes associated with energy metabolism, suggesting an even higher energy demand for this subtype (Supplementary Fig. 3f). Hence, these data suggest that Ia$_3$-PNs represent a muscle-specific Ia-PN subtype with higher dynamic intrinsic properties and muscle-specific contribution to MSs sensory feedback.

Although all II-PN subpopulations exhibited generally smaller soma size compared with Ia- and Ib-PNs, the II$_3$-PNs were on average larger (Fig. 3i), suggesting that they might have the fastest conduction velocity among II-PNs. Anatomically, II$_1$-, II$_3$- and II$_4$-PNs were all found in brachial and lumbar DRG that innervate mostly the limbs and some axial muscles (Fig. 3j). In contrast, II$_2$-PNs were proportionally more abundant in the thoracic DRG that supply only axial muscles (Fig. 3j). We found that 46.8 ± 11.9% and only 11.4 ± 3.6% of II$_2$-PNs were *Fxyd7*[+] in brachial and thoracic DRG, respectively, suggesting that II$_2$-PNs might specifically innervate axial and intercostal muscles[22].

Altogether, these data extended our current view on the modular organization of proprioceptive feedback and revealed 8 functional subtypes of PNs with distinguishable molecular profiles, spatial distributions (Fig. 3k), and connectivity patterns, suggesting another level of organization logic of the proprioceptive feedback.

### Molecular signatures of neuronal communication pattern in PN subtypes. 
In addition to the anatomical distribution and connectivity, the transmission of the muscle proprioceptive feedback to the central motor circuits relies on distinct operational components that define their neurotransmission mode and functionality. Using differential expression analysis, we have identified numerous genes that were specifically expressed or enriched in each PN subtype (Supplementary Dataset 1). We further categorized those genes according to their functions and, while further analysis will be necessary to confirm their expression in situ, this list of candidate genes could serve as a basis for understanding the different properties of PN subtypes. Many genes for general cellular functions, including cytoskeleton, adhesion molecules, transcription factors, cell-to-cell signaling, and neuropeptides/hormones and receptors were differentially expressed amongst PNs, and are presented in Supplementary Fig. 4a–e. We next analyzed expression of genes, which belong to functional categories that characterize the physiological properties of mechanosensory neurons, including mechanosensitive ion channels (MSICs), integrin-mediated mechanotransduction, voltage-gated ion channels (VGICs) and neurotransmitter transport and receptors (Fig. 4a–f). While many MSICs, including *Piezo2*[23], were found equally expressed in all PN subtypes (Fig. 4b), we identified six MSICs that were differentially expressed among them, indicating that each subtype of PNs

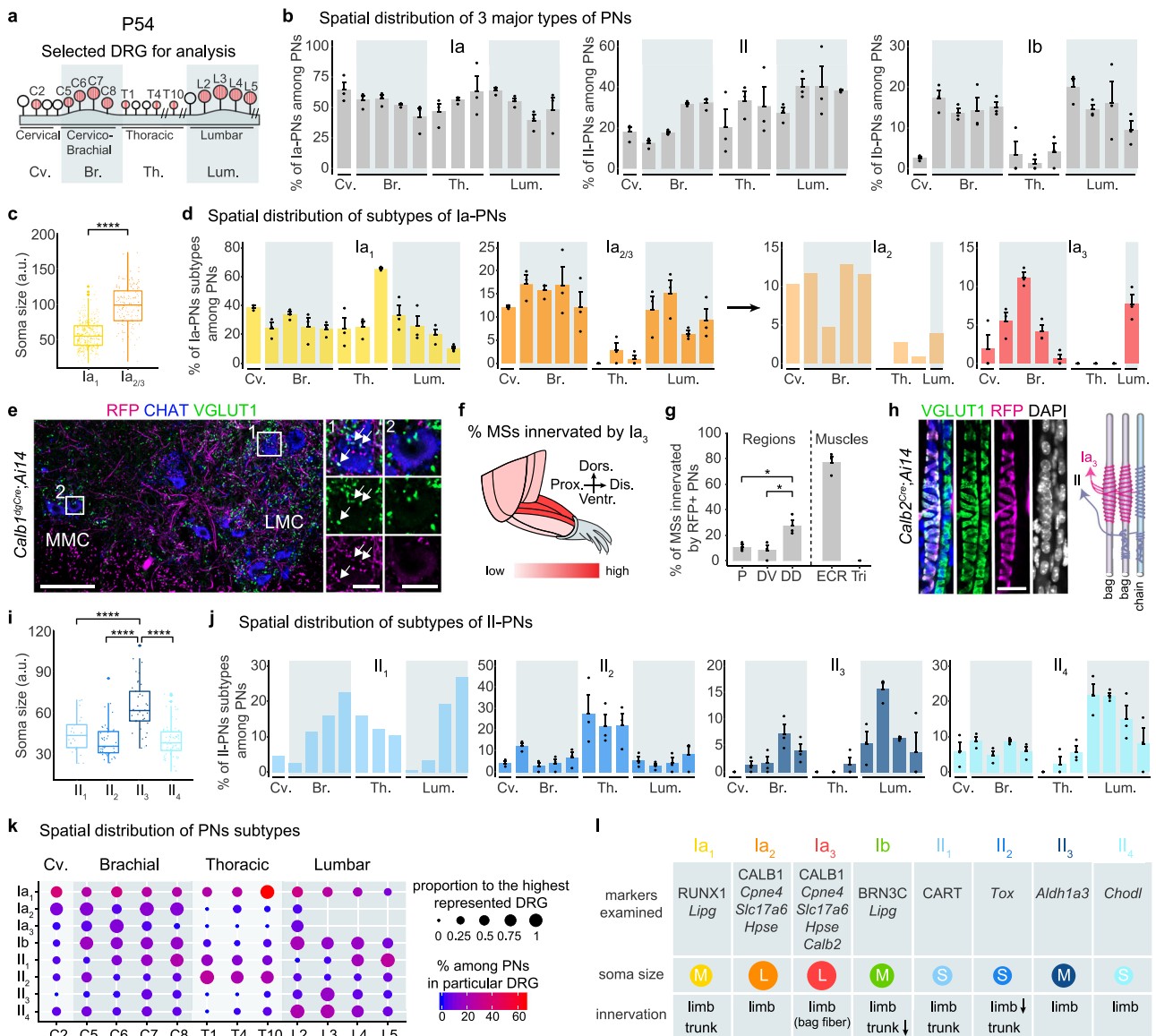

**Fig. 3 Functional subdivisions of PNs. a** Schematic depiction of the selected DRG (highlighted in red) to study the spatial distribution of PNs along the rostral-caudal axis of the trunk. **b** Spatial distribution of the three major types of PNs in representative DRG (see A) (n = 3 animals). Data are presented as mean ± SEM; dots represent values from individual animals. See "Methods" for details. **c** Comparison of cross-nucleus soma sizes of Ia-PNs subtypes depicted in box and whisker plot. Lower and upper hinges: first and third quartiles; the horizontal line: median; the whiskers extend to the value no further than 1.5 * IQR from the hinge; large dots: outliers. Each small dot represents a cell: Ia$_1$ (n = 342 cells), Ia$_2$ and Ia$_3$ together (n = 148 cells). Two-tailed t-test, ****p < 0.0001. **d** Spatial distribution of Ia-PNs subtypes in representative DRG (see A). Data are presented as mean ± SEM (n = 3 animals); dots represent values from individual animals. See Methods for details. **e** Ventral spinal cord section of P54 Calb1$^{dgCre}$;Ai14 mice labeled for CHAT (motor neurons) and VGLUT1 (sensory terminals), showing RFP$^+$ sensory boutons onto motor neurons in LMC but not MMC. Scale bar: 100 μm. Scale bars of the micrographs: 20 μm. **f** Schematic depiction of the over-representation of Ia$_3$-PNs innervation in the distal-dorsal region of the mouse forelimb. **g** Proportion of MSs innervated by RFP$^+$ afferents in P30 Calb2$^{Cre}$;Ai14 mice (n = 3 animals). P: proximal; DD: distal-dorsal; DV: distal-ventral; ECR: extensor carpi radialis; Tri: triceps. Data are presented as mean ± SEM; dots represent values from individual animals. Two-tailed t-test, *p < 0.05. **h** MS of P30 Calb2$^{Cre}$;Ai14 mice stained for RFP, VGLUT1 and DAPI, showing that RFP$^+$ afferents innervate only the nuclear bag but not chain fibers. Bag and chain fibers are distinguishable by their sizes and organization of nuclei: chain fibers have nuclei aligned in a chain, while bag fibers have many nuclei stacked in bags. Right panel: schematic representing the finding. Scale bar: 20 μm. **i** Quantification of cross-nucleus soma sizes of II-PNs subtypes depicted in box and whisker plot. Lower and upper hinges: first and third quartiles; the horizontal line: median; the whiskers extend to the value no further than 1.5 * IQR from the hinge; large dots: outliers. Each small dot represents a cell: II$_1$ (n = 25 cells), II$_2$ (n = 43 cells), II$_3$ (n = 37 cells), II$_4$ (n = 65 cells). Two-tailed t-test, ****p < 0.0001. **j** Spatial distribution of II-PNs subtypes in representative DRG (see a). Data are presented as mean ± SEM (n = 3 animals); dots represent values from individual animals. See "Methods" for details. **k** Dot plot summarizing the spatial distribution of all PN subtypes. The size of the circle reflects how a subtype is distributed along the rostral-caudal axis where the presence of the subtype in each DRG is proportional to the highest represented DRG. The color intensity reflects the percentage of different subtypes among all PNs in the same DRG. **l** Table summarizing main characteristics of PN subtypes. The illustrated soma sizes of the subtypes are proportional to the average soma sizes of the respective PN subtypes observed in situ. Source data are provided as a Source Data file.

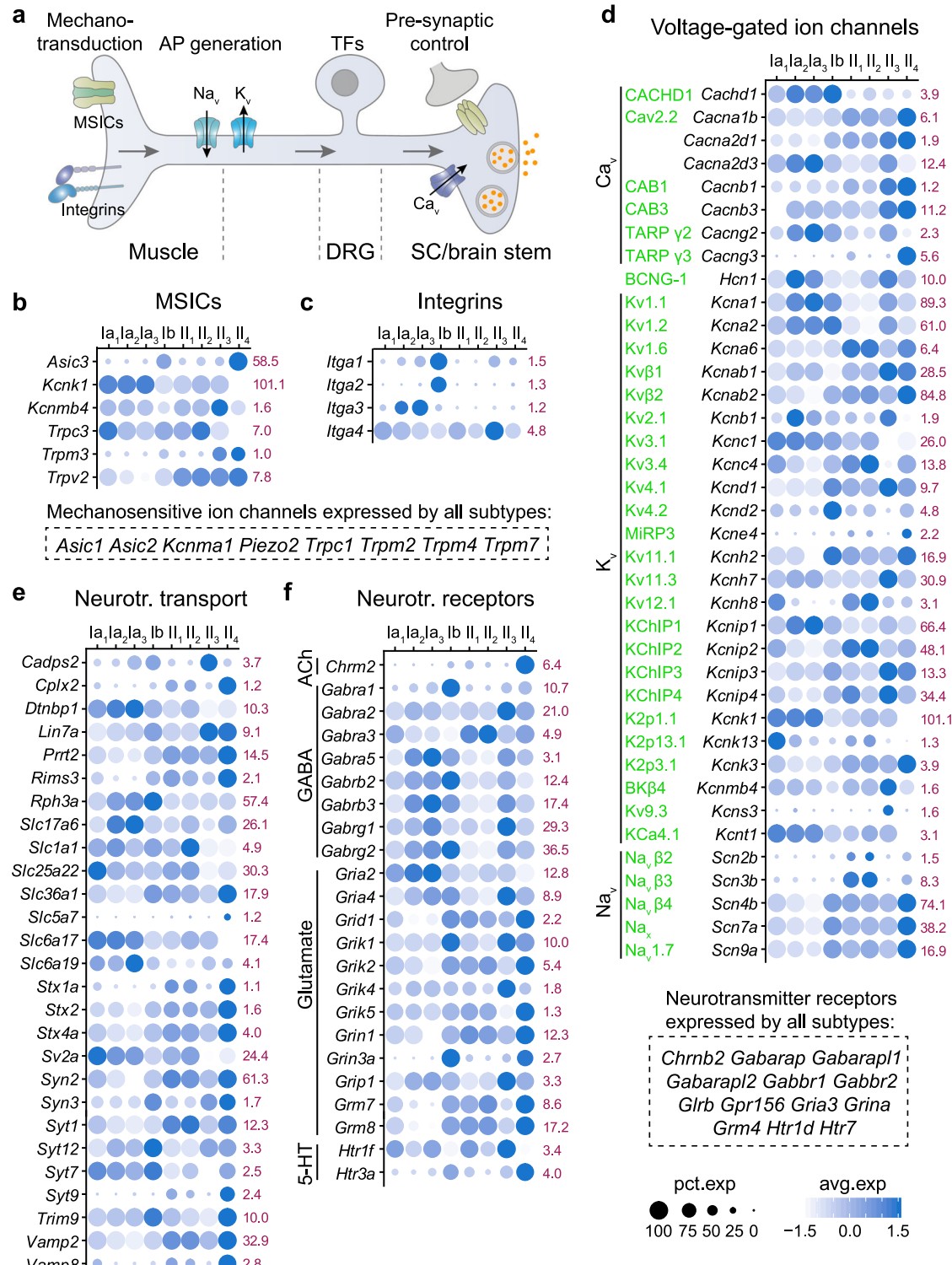

**Fig. 4 Unique molecular signatures of neuronal communication pattern in the PN subtypes. a** Schematic depicting gene categories important for intrinsic electrical properties and input-output communication patterns of PNs. **b–f** Dot plots of differentially expressed genes for functional neuronal properties, including "mechanosensitive ion channels (MSICs)", "integrins", "voltage-gated ion channels (VGICs)", "neurotransmitter transport" and "neurotransmitter receptors". The highest mean of expression among the subtypes is specified in red. Protein names for VGICs are shown in green.

might require additional MSICs to acquire their unique mechanosensory property. Interestingly, the Ib-PNs unique expression of *Itga1* and *Itga2* (integrins α1 and α2) — the only known integrin subunits to heterodimerize with β1 subunit to mediate cell surface binding to collagens[24] — suggests that the mechano-activation of Ib-PNs might specifically involve α1β1

and α2β1 integrins (Fig. 4c). In this context, CHAD (chondroadherin), the main marker of Ib-PNs, is a secreted protein that interacts with both α2β1 integrins and collagens[25] and is thus expected to mediate Ib nerve endings-matrix interactions.

Many voltage-gated sodium (Na$_v$), calcium (Ca$_v$), and potassium (K$_v$) channels were differentially expressed in PN

subtypes (Fig. 4d). Among them, $K_v$ channels and their accessory subunits exhibited the most diverse expression patterns (Fig. 4d), which would likely contribute to the regulation of various firing properties and signal integration of the different PN subtypes, as for the major types of somatosensory neurons[26]. *Slc17a6* (VGLUT2), the glutamate transporter that mediates high fidelity of neurotransmitter release[27], was highly enriched in $Ia_2$- and $Ia_3$-PNs (limb-innervating PNs) (Fig. 4e), suggesting increased synaptic strength with their postsynaptic partners, which might underlie a more effective and sensitive signal transmission required for the limb movements. Moreover, the gain and acuity of sensory inputs are modulated presynaptically by local spinal interneurons, in this context, many neurotransmitter receptors were also differentially expressed among PN subtypes with a few showing subtype-restricted expression (Fig. 4f). The observed discrete expression of neurotransmitter receptors in PNs likely contribute to controlling the sensitivity of the different proprioceptive sensory channels centrally but also to a possible discriminatory capacity of the proprioceptive system to focus on relevant afferent inputs, as shown between distinct sensory modalities[28,29].

**Postnatal emergence of PN subtypes**. The discovery of the large diversity of PN subtypes leads us to ask when and how they emerge during development. Using scRNAseq of PNs at E16.5, a stage marked by already established connections with their peripheral targets and an ongoing central axon growth process to their spinal cord output neurons, three molecularly distinct clusters were identified (Fig. 5a). However, the transcriptomic profiles of E16.5 PNs exhibited large discrepancy from P54 PNs, making it challenging to connect those two stages with computational tools. Thus, to understand the classification of E16.5 PNs, we have identified marker genes for each cluster (cl.1: *Vstm2b*; cl.2: *Tnfaip8l3*; cl.3: *Doc2b*) and used them to label the clusters in vivo (Fig. 5b, c). In order to identify Ia-PNs among the three major clusters, we injected rhodamine dextran (Rh-Dex) in the MN region of the brachial ventral spinal cord of E16.5 embryos to retrogradely target Ia-PNs (Fig. 5d–f) (see "Methods" for details). After ex vivo incubation, both pre-motor interneurons in the spinal cord and Ia-PNs in the DRG were selectively labeled by Rh-Dex (Fig. 5e, f) (~90% of Rh-Dex$^+$ DRG cells were PNs). Interestingly, while *Tnfaip8l3*$^+$, *Vstm2b*$^+$ and *Doc2b*$^+$ PNs represented 20.3, 37.0 and 30.0% of PNs in brachial DRG, respectively, the *Tnfaip8l3*$^+$ population was significantly over-represented (68.0%) in Rh-Dex$^+$ PNs (Fig. 5g), suggesting the cl.2 to be Ia-PNs. A small number of *Vstm2b*$^+$ and *Doc2b*$^+$ cells were also traced in our experiments (Fig. 5g). This could be explained by the diffusion properties of the dextran that might be taken up by few nerve endings of other PNs terminating in the vicinity of the ventral horn of the spinal cord at this early stage (see "Methods" for details), but also by the incomplete cell-type specificity of the markers used, which at this stage showed enrichment but not unique expression in a given cluster (Fig. 5b). To genetically trace the *Doc2b*$^+$ PNs lineage (cl.3), we generated a transgenic mouse driver line *Doc2b$^{ddCre}$*, expressing trimethoprim (TMP)-inducible ddCre under the control of the endogenous *Doc2b* locus and where Cre recombinase is fused to the destabilizing DHFR domain, which can be transiently stabilized by the administration of TMP (Fig. 5h)[30]. Using *Doc2b$^{ddCre}$;Ai14* mouse, we genetically traced the *Doc2b$^+$* cells from E16.5 and examined them postnatally with adult PN markers (Fig. 5i). Notably, we observed that almost 90% of Ib-PNs (BRN3C$^+$) were labeled by RFP at brachial level (Fig. 5i), while less than 30% of Ia- and II-PNs were positive, strongly suggesting that *Doc2b* marked mostly the Ib-PNs at E16.5. The small fraction of traced

Ia- or II- PNs in *Doc2b$^{ddCre}$;Ai14* mouse might be the result of a baseline recombination often seen in *ddCre* mouse lines (see Methods for details). Additionally, at E16.5, *Doc2b$^+$* PNs were found specifically enriched in DRG innervating the limbs (Fig. 5j), another characteristic of Ib-PNs (Fig. 3b). Together, these data suggest that cl.2 and cl.3 of E16.5 PNs identify Ia- and Ib-PNs, respectively, and the remaining cl.1 thus most likely represented II-PNs (Fig. 5k). We have also provided a list of the transcription factors expressed in each PN type at this critical developmental stage (Supplementary Fig. 5a), which will help future investigations of the molecular programs that control the differentiation and connectivity patterns of PNs during early development.

While the three majors clusters could be split further into subclusters, those were only found at this early developmental stage and a gene ontology analysis of these subdivisions' markers revealed a high enrichment in terms involved in synapse assembly, cell adhesion and axon guidance. This suggests that the upregulation and differential expression of genes necessary for the assembly of sensorimotor circuit around E16.5 might generate transcriptional diversity resulting in transient subdivisions amongst the major cell types, as recently suggested for the developing visual system in *drosophila*[31]. Further developmental studies will be necessary to explain the biological basis of the subcluster diversity of PNs at this stage (Supplementary Fig. 5b, c).

We then asked whether PNs diversification begin during the early postnatal period, when activity-dependent mechanisms remodel neural circuits[32,33]. For this, we generated scRNAseq data of P5 PNs, when the pups still exhibit immature locomotor behavior, and identified 6 molecularly distinct clusters at this stage (Fig. 5l). The major PN types Ia (cl.1), Ib (cl.2) and II (cl.3,4,5,6) were already identified by their overall transcriptomic similarity to adult PN types and expression of adult markers (Fig. 5l and Supplementary Fig. 6a, b). Interestingly, II-PN subtypes were already present at this stage and expressed corresponding adult markers (Fig. 5m), while Ia-PNs were not yet clearly subdivided into the subtypes found in adult (Fig. 5m). In addition, P5 Ia-PNs expressed many Ib- or II-PN markers, while those were reduced or absent in adult Ia-PNs (Supplementary Fig. 6c). Specifically, many genes related to neuronal functions (e.g., ion channels and neurotransmitter receptors), which were expressed in P5 Ia-PNs were later downregulated at P54 (Supplementary Fig. 6d), likely reflecting that the subtype characters of Ia-PNs might be sharpened through repressing genes that are nonspecific or no longer required for their mature neuronal activities.

Taken together, these observations suggest that the three main types of PNs (Ia, Ib, and II) differentiate simultaneously with the formation of the sensorimotor circuit during embryogenesis (Fig. 5k), while PN subtypes emerge postnatally (Fig. 5n). The diversification of the II-PNs into four subtypes around birth could result from changes in the environment, possibly including motor or spontaneous activity. Ia-PNs diversification into three subtypes might require further sensory experience through patterned motor activity and involve trimming away nonspecific neurotransmission-related genes of the alternative adult PN subtypes.

**Activity-driven plasticity of adult Ia-PNs**. The observation of the relatively late differentiation of Ia-PNs led us to investigate whether the identity of Ia-PN subtypes remains versatile in adult life following changes in sensorimotor activity. To test this, we used motor skill learning that has been intensely studied in the realm of neuroplasticity[34–36]. More specifically, we assessed the composition of Ia-PN subtypes in DRG of wild-type mice with

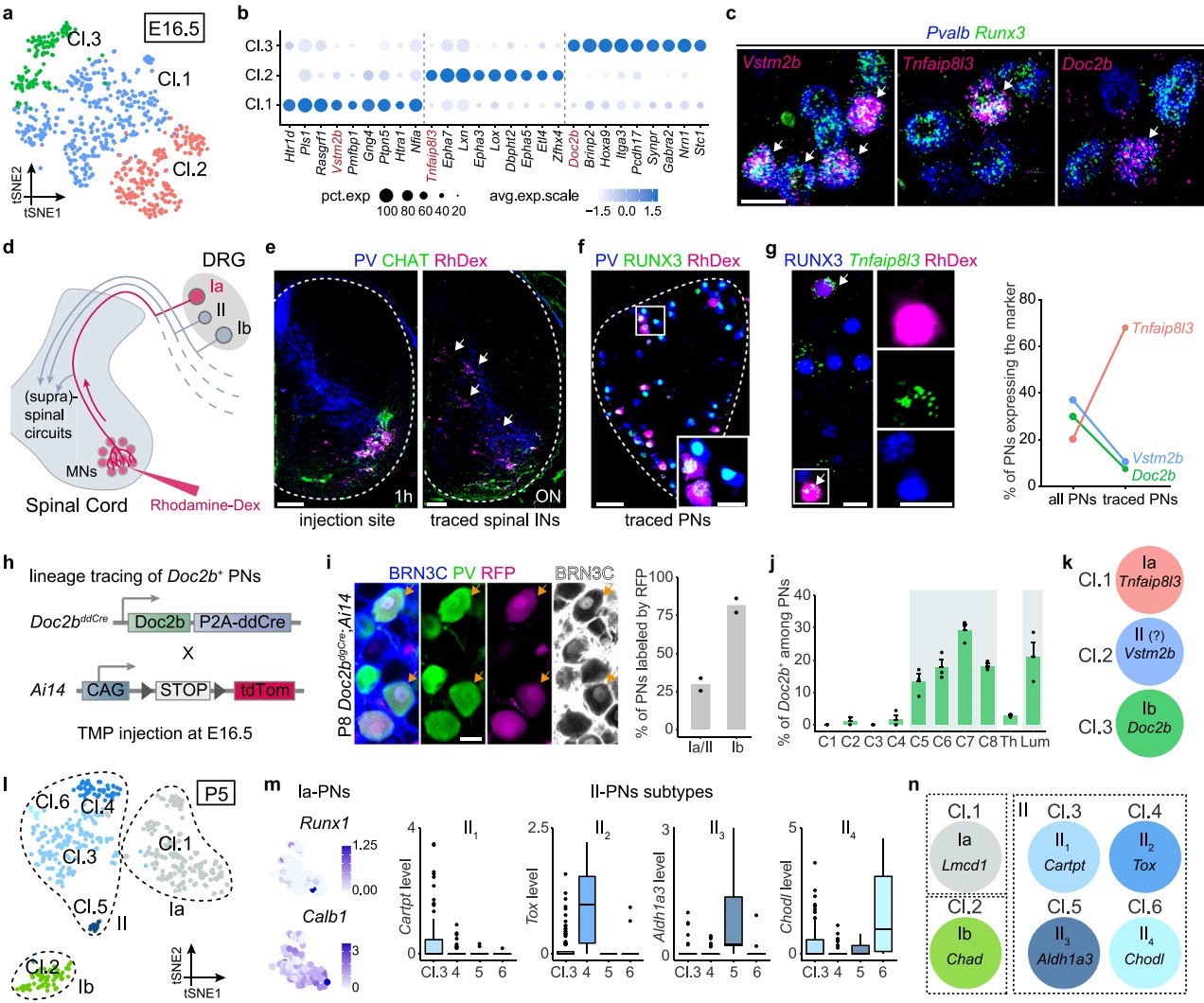

**Fig. 5 Emergence of PN subtypes during development. a** Three molecularly distinct clusters of E16.5 PNs shown by tSNE. **b** Dot plot illustrating examples of marker genes in each cluster. **c** In vivo validation of the three E16.5 PN populations by RNAscope using the identified markers (magenta) and PN markers *Pvalb* and *Runx3*. Scale bar: 20 μm. **d** Strategy to retrogradely label Ia-PNs in E16.5 embryos. **e** Left: transverse section of spinal cord 1 h post-injection stained for PV and CHAT, showing the injection site in the motor neurons area. Right: transverse section of spinal cord over night (ON) post-injection stained for PV and CHAT, showing the retrogradely labeled pre-motor spinal interneurons. Scale bars: 100 μm. **f** DRG section from embryo 6 h post-injection stained for PN markers PV and RUNX3, showing the retrogradely labeled Ia-PNs. Scale bar: 50 μm. Scale bar of the micrograph: 20 μm. **g** Retrogradely labeled DRG section stained for PN marker RUNX3 (immunostaining) and subtype marker *Tnfaip8l3* (RNAscope). Right: Proportion of the three PN populations among all PNs (*Pvalb+*/*Runx3+*) and retrogradely traced PNs from experiment in **d** (n = 4 embryos for *Tnfaipl3* and *Doc2b*, n = 3 embryos for *Vstm2b*). Scale bar: 20 μm. Scale bar of the micrograph: 20 μm. **h** Strategy to genetically trace the *Doc2b+* PNs lineage (TMP: trimethoprim). **i** P8 DRG section from *Doc2b^{ddCre}*;*Ai14* mice injected with TMP at E16.5. The sections were stained for PN marker PV and postnatal Ib-PN marker BRN3C. The staining of BRN3C in gray scale is pseudo-coloring from blue. Scale bar: 20 μm. Right: proportion of Ia/II-PNs and Ib-PNs labeled by RFP, respectively, presenting enriched tracing of Ib-PNs (n = 2 animals); dots represent values from individual animals. **j** Spatial distribution of *Doc2b+* PNs in representative DRG of E16.5 embryos. Data are presented as mean ± SEM (n = 3 animals). Data for Th. and Lum. regions represent an average of the proportion observed in segments T2-T12 and L1-L5, respectively. **k** Correspondence between PN types and the clusters identified by scRNAseq at E16.5. **l** tSNE of PNs depicting molecularly distinct clusters at P5. **m** Box and whisker plots showing the marker expression of 4 II-PN subtypes at P5, while Ia-PN subtypes were indistinguishable by their subtype markers at this stage (e.g., *Runx1*, *Calb1*). Lower and upper hinges: first and third quartiles; the horizontal line: median; the whiskers extend to the value no further than 1.5 * IQR from the hinge; large dots: outliers. For the box plots, Cl.3 (n = 108), Cl.4 (n = 96), Cl.5 (n = 14), Cl.6 (n = 11). **n** Correspondence between PN types and the clusters identified by scRNAseq at P5. Source data are provided as a Source Data file.

access to free-wheel running for 4 weeks, which is known to induce experience-dependent plasticity in the central nervous system[36–38]. During the entire training period, running (Run) and sedentary (Sed) mice were single housed in enhanced environment: large rat cage, wooden sticks, nest and free access to a running wheel (Fig. 6a). The wheel was in locked mode in the sedentary condition (Fig. 6a). Each exercised mouse spent substantial amount of time on the wheel and ran on average 8 km/

day. No gain of weight in the running group compared with the sedentary groups showed the efficiency of running (Fig. 6b). At the end of the training period, mice were sacrificed and analyzed for PNs phenotype. The overall proportion of Ia-PNs (among all PNs) (Fig. 6c) and their total number (raw counts, normalized to sedentary condition: Sed 1 ± 0.13 and Run 0.98 ± 0.005) in brachial DRG of sedentary and running animals were identical. Importantly, in the running condition we observed a decrease in

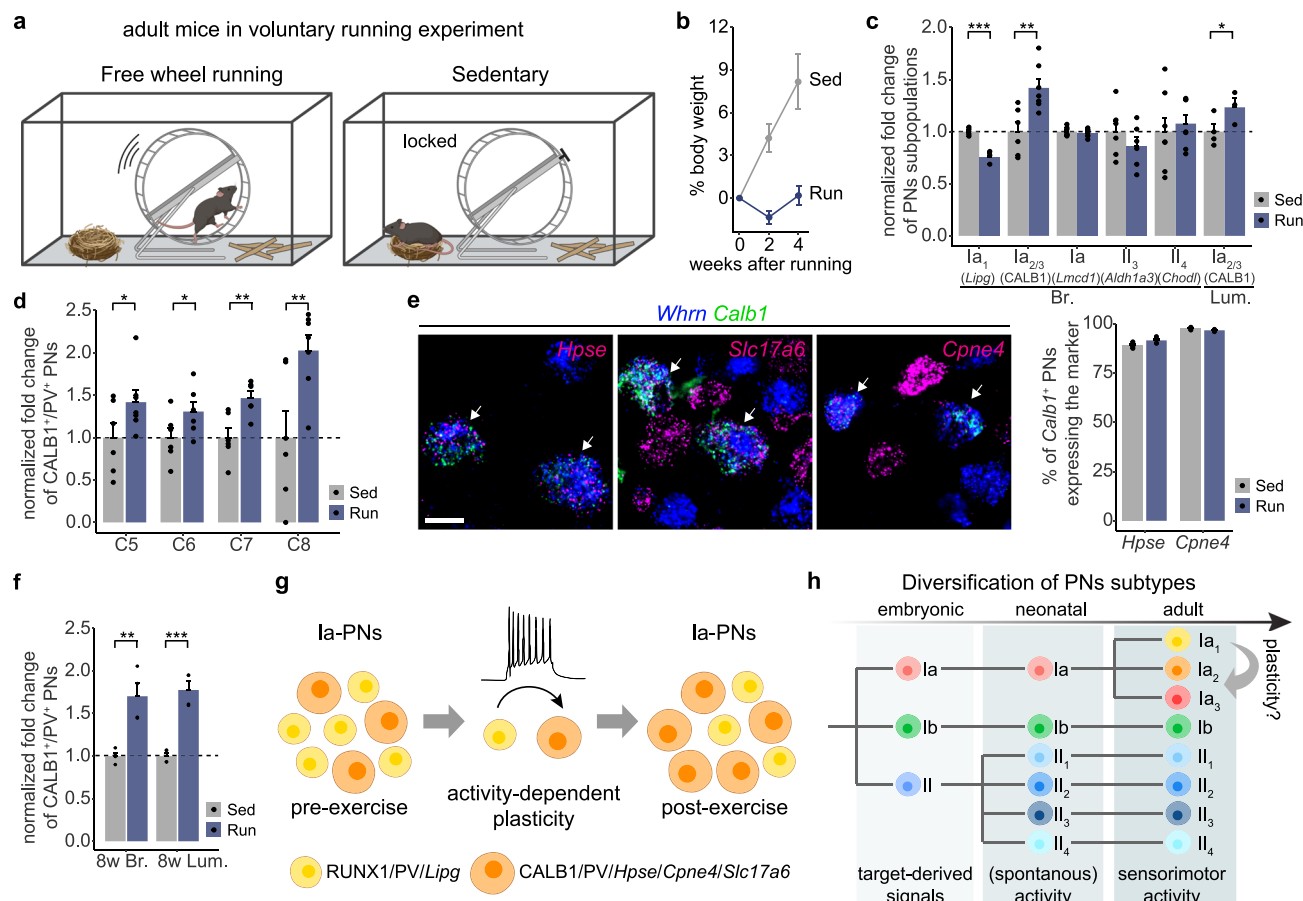

**Fig. 6 Activity-driven plasticity of adult Ia-PN subtypes. a** Schematic depiction of the housing environment during the voluntary running experiment.
**b** Percentage of weight change in sedentary (Sed) and free wheel running (Run) groups. Data are presented as mean ± SEM ($n = 7$ animals for both groups). **c** Fold change of proportion of PN subtypes in 4-week running group compared with sedentary group: $Lipg^+/Whrn^+$ cells (Ia$_1$) among $Whrn^+$ cells, CALB1$^+$/PV$^+$ cells (Ia$_{2/3}$) among PV$^+$ cells, and $Lmcd1^+/Whrn^+$ (Ia), $Aldh1a3^+/Whrn^+$ (II$_3$), and $Chodl^+/Whrn^+$ (II$_4$) cells among $Whrn^+$ cells. Ia$_1$: $n = 4$ animals for sedentary group and $n = 3$ animals for running group; Ia$_{2/3}$ (Br.): $n = 6$ animals for sedentary group, $n = 7$ animals for running group; Ia: $n = 7$ animals for both groups; II$_3$: $n = 6$ animals for both groups; II$_4$: $n = 7$ animals for both groups; Ia$_{2/3}$ (Lum.): $n = 4$ animals for sedentary group, $n = 3$ animals for running group. Data are presented as mean ± SEM; dots represent values from individual animals. One-tailed $t$-test, *$p < 0.05$, **$p < 0.01$, ***$p < 0.001$.
**d** Fold change of proportion of CALB1$^+$/PV$^+$ PNs in individual brachial segments in 4-week running group compared with sedentary group ($n = 7$ animals). Data are presented as mean ± SEM; dots represent values from individual animals. One-tailed $t$-test, *$p < 0.05$, **$p < 0.01$. **e** RNAscope images showing the expression of other Ia$_{2/3}$-PN markers (magenta) in $Calb1^+$ PNs ($Whrn^+$) on the brachial DRG sections of the running mice. Scale bar: 20 μm. Right: percentage of $Calb1^+$ PNs expressing other Ia$_{2/3}$ markers in sedentary and running groups, respectively ($n = 4$ animals). Data are presented as mean ± SEM; dots represent values from individual animals. **f** Fold change of proportion of CALB1$^+$/PV$^+$ PNs in 8-week running group compared with sedentary group ($n = 4$ animals). Data are presented as mean ± SEM; dots represent values from individual animals. One-tailed $t$-test, **$p < 0.01$, ***$p < 0.001$.
**g** Schematic summarizing the results from the voluntary running experiment. **h** Model of the diversification of PN subtypes. Br: brachial DRG (C5-8); Lum: lumbar DRG (L2-5). Source data are provided as a Source Data file.

the proportion of $Lipg^+/Whrn^+$ PNs (labeling Ia$_1$-PNs) and an increase in CALB1$^+$/PV$^+$ PNs (labeling Ia$_{2/3}$-PNs) at limb level (Fig. 6c, d). In contrast, analysis of II-PNs in DRG from the same animals did not reveal any change in subtype composition (Fig. 6c), suggesting that this type of neuronal plasticity mediated by increased locomotor activity is specific to Ia-PNs. To confirm the observed changes in Ia-PNs subtype composition, we tested other Ia$_{2/3}$-PN markers such as $Hpse$, $Slc17a6$ and $Cpne4$. In the running mice, virtually all $Calb1^+$ PNs were found to also express these Ia$_{2/3}$-PN markers (Fig. 6e), indicating that the transcriptional dynamic in Ia-PNs following sustained exercise training is not limited to a single-marker gene. After 8-week training, the proportional increase of Ia$_{2/3}$-PN markers was further enlarged to 70 and 77% at brachial and lumbar level, respectively (Fig. 6f). Altogether, these data suggest an increase in a Ia$_{2/3}$-PN identity after running exercise at the expense of a Ia$_1$-PN identity (Fig. 6g). As described earlier, among PNs, the Ia$_{2/3}$ subtypes

would represent highly dynamic populations. Thus, the observed plasticity of Ia-PNs with versatile subtype composition might serve as one mechanism of the adult proprioceptive system to adapt to changing motor behavior. Whether this plasticity is controlled by activity-dependent programs or target-derived or systemic factors, e.g., cytokines or growth factors (from muscles or blood), is awaiting to be addressed.

## Discussion

The first main insight from our results is the demonstration of an extensive diversity of PNs with discrete anatomical organization and unique genetic identities, a molecular and cellular diversity long searched for but never observed. This classification of PNs suggests that the proprioceptive system is endowed with a vast array of PN subtypes that are necessary to extract and transmit the various features of muscle activities across the body. The

second main result is the late emergence of the dynamic subtypes of the Ia-PNs and their plasticity in the adult in the form of changing subtype identity after sustained exercise training. This plasticity suggests neuronal individuality in the nervous system to adapt its performance to changing environment.

Adult PNs have long been recognized to comprise three functional types (Ia-, Ib-, and II-PNs) based on their anatomy and electrophysiological properties ex vivo, however, molecular tools to distinguish them have been lacking, preventing research into their individual functions in vivo, especially their acute modulation in adult. In this study, we identified eight discrete subtypes among the three major types, with their respective molecular markers, hence providing tools for direct (intersectional) genetic strategies to address their functional connectivity and roles in vivo. Importantly, our comprehensive molecular delineation of PN subtypes also provides a large set of functional genes that are predictive of the biophysical and physiological features of PNs and will stimulate future investigations in the field.

The current view on the organization of the proprioceptive feedback describes an elementary modular organization, where three types of PNs each relay the muscle length, speed and force, respectively, omitting the fact that muscles are distinct in their functionality and that their activities cover a diverse range of sensitivity. Thus, our discovery of the eight functional subtypes of PNs extend this view and demonstrate a more sophisticated organization where distinct PN subtypes serve only selective muscle targets (Fig. 3l), and would differ in their dynamic sensitivity, which would increase for instance from Ia$_1$- to Ia$_3$-PNs. These differences could participate in the detection and transmission of a large array of sensory stimuli. How these various sensory inputs are integrated and processed centrally for controlling motor outputs are intriguing topics for future studies, which can now be envisaged using our molecular tools. Moreover, considering the executive role of the proprioceptive feedback in directing restoration of the motor function after spinal cord injury[2], our findings will also help developing experimental strategies to study and modulate specific MS feedback channels for therapeutic intervention.

Our data indicate that diversification into these PN subtypes follows a three-step model, where an early genetic programming establishes PN cardinal identity (after neurogenesis)[39,40] that is refined into PNs types following innervation[41], and which later diversify into several subtypes postnatally (Fig. 6h). The observation that the diverse PN subtypes found in adult could be identified only postnatally favors the influence of neuronal activity in driving their later diversification program[42,43]. Moreover, the observed delayed diversification of Ia-PN subtypes suggests experience-dependent plasticity involving a tight interplay between sensorimotor experience and neuronal identity. Remarkably, we further demonstrate that the substantial plasticity of Ia-PN subtypes persist in the adult by changing subtype identity following sustained running paradigm that has been associated with motor skill learning[36]. This reveals another level of neuroplasticity given that most studies on (non-pathological) neuroplasticity in adult has been focusing on cortical and subcortical areas and demonstrated structural adaptations and/or local changes in synaptic strength. Future studies will be necessary to understand the physiological relevance of such plasticity and whether the observed changes reflect a real switch in cell type and if it is reversible or remains stable over time. It will also be interesting to address the influence of the environment in generating neuronal individuality in other regions of the peripheral nervous system, and how this heterogeneity might correlate with performance in sensory processing functions and eventually individual behavioral differences.

## Methods

**Mouse lines**. All animal experiments were approved by the local ethical committee (Stockholms Norra djurförsöksetiska nämnd, Sweden) and conducted following the ethical guidelines described in the *Swedish Animal Agency's Provisions and Guidelines for Animals Experimentation Recommendations*. Mice were housed in cages in groups, with food and water *ad libitum*, under 12 h light–dark cycle conditions. *PV$^{Cre}$;Ai14* mice were crossed from *PV$^{Cre}$* (from The Jackson Laboratory[44]) and *Ai14* (from The Jackson Laboratory[45]), and used to genetically label PNs for scRNAseq experiments. *Egr3$^{WGA}$* mice[19] were used to specifically label MSs-innervating PNs. *ChAT$^{Cre}$;RGT* mice were crossed from *ChAT$^{Cre}$* (from Ole Kiehn lab[46]) and *RGT* (from The Jackson Laboratory[20]), and used to specifically express avian receptor protein TVA and rabies glycoprotein in MNs for rabies virus infection. *Calb2$^{Cre}$;Ai14* mice[47] were used to genetically label *Calb2$^+$* cells in DRG for neuroanatomical tracing. *Calb1$^{dgCre}$;Ai14* mice were crossed from *Calb1$^{dgCre}$* (from The Jackson Laboratory[48]) and *Ai14*, and used to genetically label and trace the nerve endings of adult *Calb1$^+$* PNs. *Doc2b$^{ddCre}$;Ai14* mice were crossed from *Doc2b$^{ddCre}$* (generated in collaboration with Gurumurthy lab) and *Ai14*, and used for lineage tracing of embryonic *Doc2b$^+$* PNs. *C57BL/6J* mice were received from The Jackson Laboratory (stock #000664) and used for most experiments unless otherwise specified. Primers used for genotyping of the mouse strains are listed in Supplementary Table 1.

**Generation of *Doc2b$^{ddCre}$* mouse model**. The *Doc2b$^{ddCre}$* mouse model was created using the *Easi*-CRISPR method[49,50]. Briefly, a long single-stranded DNA donor containing P2A-ddCre cassette was used as a repair template to insert immediately before the stop codon of *Doc2b* gene. The donor DNA was injected into pronuclei derived from *C57BL6/J* mice along with CRISPR Ribonucleoproteins (a synthetic guide RNA and Cas9 protein mix). The final concentrations of components in ctRNP (crRNA + tracrRNA) preparations were 10 ng/μl of guide RNA and 10 ng/μl of Cas9 protein. The ssDNA donor was mixed with ctRNP complexes at final concentration 10 ng/μl and the final injection mixes were passed through Millipore Centrifugal Filter units (EMD Millipore, cat#UFC30VV25) and spun at $21,000 \times g$ for 5 min at room temperature. Superovulation, zygote isolation, pronuclear microinjection, surgical transfer of the zygotes into pseudo-pregnant mice were performed following the standard protocols described previously[50]. The live offspring were genotyped using 5' and 3' Junction PCRs, standard strategies used for identifying knock-in cassettes[50]. The entire region of the knock-in cassette was verified by Sanger sequencing. The model was generated at the mouse genome engineering core facility, university of Nebraska medical center. The line was shipped to Karolinska Institutet where it was rederived in *C57BL/6J* mice using in vitro fertilization.

**Trimethoprim treatment**. On the day of intraperitoneal (IP) injection, Trimethoprim (TMP) (Sigma-Aldrich, cat#T7883) was dissolved in dimethyl sulfoxide (Fisher Scientific, cat#10206581) at concentration of 50 mg/ml and diluted 1:3 into 0.9% saline solution to obtain working solution. For *Calb1$^{dgCre}$;Ai14* strain, P30 mice were treated with TMP (100 μg/g b.w.) on 3 consecutive days. Two weeks after the injection, the mice were anesthetized by isoflurane and perfused for tissue collection. For the tracing of *Doc2b$^+$* cells, pregnant females were treated with single dose (125 μg/g b.w.) of TMP at E16.5. The P8 pups were anesthetized by isoflurane and perfused for tissue collection. Owing to the nature the TMP-inducible *ddCre* mouse lines, it is possible that for some genes that are highly expressed, an excess amount of destabilized Cre is produced that exhausts the degeneration capacity, leading to certain level of recombination in the absence of TMP[26]. On account of this, we have observed that in the absence of TMP *Doc2b$^{ddCre}$;Ai14* mice displayed recombination in 13.5% of PNs, among which the proportion of *Lmcd1$^+$/Whrn$^+$*, *Fxyd7$^+$/Whrn$^+$* and *Chad$^+$/Whrn$^+$* was 43.5%, 29.0% and 29.5%, respectively.

**Retrograde labeling of Ia-PNs**. For the retrograde labeling of adult Ia-PNs, *ChAT$^{Cre}$* mice were crossed with *RGT* mice to obtain the *ChAT$^{Cre}$;RGT* strain (heterozygous for both alleles). Viral injections with EnvA-ΔG-eGFP were performed in multiple areas of the lumbar cord once the animals became adult (60 days old). For the injections, the animal was anesthetized with isoflurane and shaved on the dorsal side. Sodium iodine was applied on the shaved skin. Eye ointment was applied on the eyes to prevent de-hydration. After skin incision, the spinal column with the lumbar segments were localized (from the 12th thoracic vertebra to the 1st lumbar vertebra) and fixed with two holders on the left and right side to prevent movement (mostly due to respiration). A vertical incision was made in front of the spinous process followed by the cut of the Ligamentum Flavum between the two vertebral bodies. At this point the spinal cord appeared. This procedure was repeated three times to expose and inject three different lumbar segments along the spinal cord. Then, the EnvA-ΔG-eGFP was bilaterally injected (200 nL per segment, 0.1 μl/1 min.), using a glass pipette (tip of about 80 μm of diameter). The injection of the virus was complete in about 15 min for the three segments. Once the injections were done, the muscles and the skin were sutured. Animals were given Buprenorphine (0.1 mg/kg) and Carprofen (5 mg/kg) subcutaneously for 2 to 5 days. Infections or excessive loss of body weight in the post-surgery period was indication of experiment termination.

For the retrograde labeling of E16.5 Ia-PNs, the anterior longitudinal ligaments of the brachial region were removed to expose the ventral spinal cord, 1% Rhodamine-Dextran (Life Technologies, cat#D-3308) was injected into the ventral horn, which is innervated by the Ia-PNs. Preparations were incubated for 6 h in DMEM-F12 medium aerated with 5% $CO_2$ in 95% $O_2$ (Carbogen) before fixation. 20% of PNs ($Pvalb^+/Runx3^+$) were Rho-Dex$^+$, while 90% of Rho-Dex$^+$ cells were PNs, showing the high efficiency and specificity of using this strategy to backfill Ia-PNs.

**Immunohistochemistry and in situ hybridization (RNAscope).** Postnatal mice were transcardially perfused with 20 ml phosphate-buffered saline (PBS) and 20 ml 4% PFA. For immunohistochemistry, tissues were dissected and post-fixed in 4% paraformaldehyde (PFA, AH diagnostics, cat#sc-281692) at 4 °C according to the sizes (P14 DRG: 1 h, P54 DRG: 1.5 h, P7 spinal column: 6 h, P54 forelimb: O/N). For RNAscope experiments, tissues were always post-fixed O/N in 4% PFA at 4 °C. The tissues were then washed three times with PBS and cryoprotected in 20% (O/N at 4 °C) and 30% (O/N at 4 °C) sucrose in PBS. Then tissues were embedded in OCT compound and sectioned at 14 μm (for DRG) or 30 μm (for muscle and spinal cord).

For immunohistochemistry, the sections were air dried for 1 h at room temperature (RT). Antigen retrieval was applied for postnatal DRG and spinal cord staining by immersing the slides in pre-heated 1x target retrieval solution (Dako, cat#S1699) for 30 min. The sections were then incubated in blocking solution (2% donkey serum (Jackson ImmunoResearch, cat#017-000-121), 0.5% triton and 0.0125% sodium azide) for 20 min at RT before applying primary antibodies for 2O/N at 4 °C. The primary antibodies used were described in the key resources table. Secondary antibodies Alexa-405, −488, −555, −647 (Life Technologies) were applied at 1:250 for O/N at 4 °C. DAPI staining was performed together with the secondary antibodies. Sections were then washed 3 times with PBS and mounted with fluorescent mounting medium (Dako, cat#S3023) for imaging.

For in situ hybridization (RNAscope), the manufacturer's protocol was followed. The probes were designed by the manufacture and available from Advanced Cell Diagnostics. The following probes were used in this study: Mm-Pvalb-C2 (#421931-C2), Mm-Whrn-C4 (#511461-C4), Mm-Runx3-C3 (#451271-C3), Mm-Fxyd7-C1 (#431141), Mm-Tox-C1 (#484781), Mm-Lmcd1-C1 (#484761), Mm-Calb1-C2 (#428431-C2), Mm-Chad-C1 (#484881), Mm-Aldh1a3-C1 (#501201), Mm-Chodl-C3 (#450211-C3), Mm-Lmcd1-C2 (#484761-C2), Mm-Fxyd7-C3 (#501021-C3), Mm-Calb2-C2 (#313641-C2), Mm-Tnfaip8l3-C1 (#484541), Mm-Vstm2b-C1 (#484861), Mm-Doc2b-C1 (#484791), Mm-Hpse-C1 (#412251), Mm-Cpne4-C1 (#474721), Hs-Calb1-C2 (#422161-C2), Mm-Slc17a6-C1 (#319171), Mm-Lipg-C1 (#492521).

**Primary antibodies.** Rabbit anti-WHRN (from Joriene de Nooij lab), mouse anti-ISLET1 (DSHB, cat#39.4D5), chicken anti-RFP (Rockland, cat#600-901-379S), rabbit anti-VGLUT1 (SYSY, cat#135303), guinea pig anti-VGLUT1 (SYSY, cat#135304), rabbit anti-FXYD7 (Sigma-Aldrich, cat#HPA026916), rabbit anti-LMCD1 (Human Protein Atlas, cat#HPA024059), goat anti-WGA (Vector Laboratories, cat#AS-2024), mouse anti-BRN3C (Santa Cruz Biotechnology, cat#sc-81980), goat anti-PV (Swant, cat#PVG-213), rabbit anti-PV (Swant, cat#PV27), rabbit anti-CART (from Igor Adameyko lab), rabbit anti-RUNX1 (from Thomas Jessell lab), rabbit anti-CALB1 (Swant, cat#CB-38a), goat anti-CHAT (Millipore, cat#AB144p), DAPI (Invitrogen, cat#D1306)

**Image acquisition and analysis.** Images were acquired using Zeiss confocal microscope LSM700, LSM800, LSM880, and LSM800 airy equipped with 5x, 10x, 20x, and 40x objectives.

*Measurement of soma sizes of PN subtypes.* The soma areas of PNs were outlined using freehand selections tool in Fiji and only cells with clear nucleus with DAPI staining were considered. RNAscope was performed for canonical PN markers *Pvalb/Runx3* and subtype markers *Lmcd1, Chad, Fxyd7, Tox, Aldh1a3,* and *Chodl* to label Ia-, Ib-, II-, II$_2$-, II$_3$-, and II$_4$-PNs, respectively. Cell bodies were outlined as visualized by *Pvalb* expression, which fill in the entire cytoplasm and nucleus. Immunostaining was performed for PNs marker PVALB and subtype markers RUNX1, CALB1, and CARTPT to label Ia$_1$, Ia$_{2/3}$, and II$_1$, respectively, and cell bodies were outlined as visualized by PVALB expression.

*Quantification of PN subtypes in Fig. 3d, j, Fig. 5j, and Fig. 6c–f.* In Fig. 3d, the proportion of Ia-PNs is quantified from PNs (PV$^+$) expressing subtype marker RUNX1 (Ia$_1$-PNs) and CALB1 (Ia$_{2/3}$-PNs) using immunostaining ($n = 3$ animals). Though *Runx1* is statistically enriched in Ia$_1$-PNs, *Runx1* RNA is also lowly expressed in other PN subtypes. However, we observed that RUNX1 as protein is detected in a specific population of PNs with distinct soma size and spatial distribution, which highly suggest that it labels predominantly Ia$_1$-PNs. The proportion of Ia$_3$-PNs is quantified from PNs (*Whrn*$^+$) labeled with RFP using RNAscope in P30 *Calb2$^{Cre}$;Ai14* mice. The proportion of Ia$_2$-PNs is the deduction of Ia$_3$-PNs from Ia$_{2/3}$-PNs. In Fig. 3j, the proportion of II-PN subtypes is quantified from PNs (*Pvalb$^+$/Runx3$^+$*) expressing subtype markers *Tox* (II$_2$-PNs), *Aldh1a3* (II$_3$-PNs) and *Chodl* (II$_4$-PNs) using RNAscope ($n = 3$ animals). The proportion of

II$_1$-PNs is the deduction of II$_2$-, II$_3$-, and II$_4$-PNs from II-PNs. In Fig. 5g, j, the proportion of *Vstm2b$^+$* and *Doc2b$^+$* PNs is quantified from PNs (*Pvalb$^+$/Runx3$^+$*) expressing high level of *Vstm2b* and *Doc2b* to represent the distribution of Cl.1 and Cl.3, respectively, since *Vstm2b* and *Doc2b* is also lowly expressed by cells from other clusters. In Fig. 6c–f, in total, 4–6 sections representing each DRG were used per animal for each immuostaining or RNAscope experiment.

*Box and whisker plot.* The lower and upper hinges represent the first and third quartiles, respectively, so the box spans the inter-quartile range (IQR). The horizontal line inside the box corresponds to median. The upper whiskers extend to the largest value no further than 1.5 * IQR from the upper hinge, while the lower whiskers extend to the smallest value at most 1.5 * IQR from the lower hinge. Large dots represent the outliers beyond the range of whiskers.

**Free wheel running experiment.** Eight-week-old *C57BL/6J* male mice were single-housed and randomly assigned into two groups: sedentary (sed) and free wheel running (run). Mice assigned to the free wheel running group were given access to running wheels (Med Associates Inc, cat#ENV-047) with a counter that monitored revolutions during the experimental period (all animals ran at least 3 km/day). Mice assigned to the sedentary group were housed in similar cages with locked running wheels for the same period. At the end of the experimental period, the running wheels were removed and animals were sacrificed about 24–48 h later for tissues collection.

**Statistics and reproducibility.** Statistical data analysis was performed with Microsoft Excel, presented as mean ± standard error of mean and statistical significance was calculated using student t-tests. Degree of significance was represented as following: *$p \leq 0.05$, **$p \leq 0.01$, ***$p \leq 0.001$, ****$p \leq 0.0001$. No animals or data points were excluded from the analysis. Our sample sizes are similar to those generally employed in the field. For scRNAseq experiments, replicates were done for all stages: E16.5: 3 plates; P5: 2 plates; P54: 4 plates. For immunostaining and RNAscope experiments, replicates were performed in often $n \geq 3$ animals even in cases where only representative images were presented in the figure.

**Single-cell isolation for scRNAseq.** The same dissociation protocol was used for DRG of E16.5, P5, and P54 $PV^{Cre}$;Ai14 mice and 3 months old $ChAT^{Cre}$;RGT mice after rabies virus infection. At each stage, DRG were dissected, cut into halves (for postnatal DRGs), and collected in L-15 medium (Life Technologies, cat#21083027) on ice. The collated DRG were incubated in 2 ml DMEM-F12 medium (Life Technologies, cat#11039-021) containing 12.5 mg collagenase IV (Life Technologies, cat#17104019) and 0.8 U dispase (30 min at 37 °C) then transferred into 1 ml 0.05% trypsin solution (15 min at 37 °C) (Life Technologies, cat#25300054), and 1 ml fetal bovine serum (Life Technologies, cat#10082147) was added at the end to stop the enzymatic reaction. The tissue was then spun down at $400 \times g$ for 5 min and resuspended in 1 ml L-15 medium, followed by mechanical tituration using fire polished Pasteur pipettes until the solution homogenized and filtered through 70 μm cell strainer (BD Biosciences, cat#352350). Single RFP$^+$ cells were sorted by fluorescence-activated cell sorting (FACS) into individual wells containing lysis buffer in a 384-well plate. The plates were immediately placed on dry ice and stored at −80 °C before processed for Smart-seq2 protocol. For the rabies virus traced cells, single GFP$^+$ cells were picked with a robotic cell-picking setup, constructed in-house around an inverted Nikon microscope. Individual cells were sucked into a glass capillary attached to a CellTram syringe with visual control. By an automated x-y-z axis drive, the capillary tip was positioned in a well of a 48-well plate containing 3 μl of lysis buffer, into which the cell was released. After cell picking, the plates were immediately placed on dry ice and stored at −80 °C before further procedures.

**Single-cell RNA-sequencing.** Smart-Seq2 protocol was performed on single isolated cells in the lab (for the rabies virus traced PNs) or by Eukaryotic Single Cell Genomics Facility at SciLifeLab, Stockholm (for the E16.5, P5 and P54 PNs). The protocol was described previously[12,51].

**scRNAseq data analysis.** The data pre-processing (demultiplexing, annotation) was done by Eukaryotic Single Cell Genomics Facility at SciLifeLab, Stockholm using validated methods. Most of the downstream analysis of E16.5, P5, and P54 data followed the same pipeline using R package Seurat[13]. A step-to-step analysis of adult data is described below. The detailed analysis of E16.5 and P5 data is not be repeated in the "Methods" section, except for the selection of key parameters. The codes used in this study will be released through the lab Github channel.

*Filtration and normalization.* We obtained in total four 384-well plates for P54 PNs from two individual preparations (preparation 1: plate 1 and 2, preparation 2: plate 3 and 4). Subtle technical variations were observed between preparations but not plates from the same preparation. We thus created two Seurat objects representing two preparations and processed with the same filter and normalization procedures.

To remove empty wells and low-quality cells, we selected cells with at least 7500 unique genes detected for downstream analysis. Genes expressed in less than three cells were also removed from analysis. To be able to integrate the two preparations, we used a global normalization strategy incorporated in Seurat package to make the data comparable between different cells and between different Smart-seq2 runs. The gene expression measurements within each cell were scaled by a constant factor 100,000, then natural-log transformed. This generated a new gene expression matrix $y_{i,j} = \log(x_{i,j}Zj*100,000)$, where $x_{i,j}$ is the count of gene $i$ in cell $j$ and $Z_{i,j}$ is the total counts of all genes in cell $j$.

We calculated the dispersion (ratio of variation to mean) of genes and selected those highly variable genes from each preparation for the alignment of the two preparations. We also regressed out the cell–cell variation in gene expression driven by "percent.mito (the percentage of mitochondrial gene content)" using the ScaleData function.

*Data integration.* To reveal the shared sources of biological variation between the two preparations, we used the integration tool for scRNAseq data sets provided by Seurat package. This tool has shown better performance over other benchmarking alignment and batch correction tools, e.g., limma, ComBat. It comprises Canonical Correlation Analysis (CCA), which is a method of inferring relationship from two matrices. Given two groups of cells $X = (X1, X2,…, Xm)$ and $Y = (Y1, Y2,…, Yn)$, and if there were correlations among them, the analysis would determine a set of canonical variates (CVs, referred to as CCs in Seurat package), storing linear combinations of $X_i$ and $Y_j$, which have maximal correlation with each other. Using a union of most variable genes from both preparations as the input for CCA, we selected the first 18 CCs to approximate all CCs, which project both data sets into the maximally correlated subspace.

*Clustering and dimensional reduction.* In order to obtain an overview of hetero-geneity in the integrated P54 data, we ran the function RunTSNE using the first 18 CCs to visualize cells in the two-dimensional space and used the function FindClusters (resolution = 0.8) to identify clusters of cells. This resulted in ten clusters, among which three clusters were identified to be glia-contaminated PNs ($Sox10^+$), $Ret^+$, and $Ntrk2^+$ mechanoreceptors, respectively, and removed from the following analysis. The unbiased clustering analysis revealed seven clusters of molecularly distinct PNs, among which we observed that one cluster (later split into Cl.2 and Cl.3) exhibited large heterogeneity in the gene expression. Cl.3, identifiable by its specific expression of $Calb2$, also had enriched/specific expression of many other genes (Supplementary Fig. 1e and Supplementary Dataset 2). With the notion to not overlook the diversity of PNs, we listed Cl.3 as an individual cluster, which we believe could be revealed by unbiased clustering analysis through sequencing more PNs. These eight clusters were then clustered into three major clusters based on similarity of expression of highly specific markers unbiasedly identified, and later confirmed by their expression in situ and through analysis of their innervation profile.

*Differential expression analysis.* To identify differentially expressed genes in each cluster, we used "MAST" test implemented in the function FindMarkers[52]. The parameters "min.pct = 0.25" and "logfc.threshold = 0.3" are chosen to select genes expressed in more than 25% of cells and at least 0.3-fold difference (log-scale) in the given cluster. From the list of differentially expressed genes, all the genetic markers used in this study were further shortlisted based on their expression level and high specificity in the given cluster. As example, the expression of adult Ib-PNs marker $Chad$ is 2.7-fold (log-scale) higher in Ib-PNs compared with other PNs. Also, we have used a stringent cutoff for a gene expression to be considered positive and to be shown or discussed in the text, considering only the 50% most expressed genes from our dataset to be represented and discussed.

*Projection of scRNAseq data across data sets.* To unbiasedly assign virus-traced cells to P54 PNs clusters, we used the R package scmap[21] to project the scRNAseq data set of the virus-traced cells to the reference data set, the P54 scRNAseq data set. For this, similarities between each cell and all centroids of the reference data set are calculated using three different measurement: Pearson, Spearman, and cosine. The cell is then assigned to the cell type that corresponds to the highest similarity value. Only when at least two similarities agree with each other and at least one similarity score is above 0.7, the cell could be assigned. P5 PNs were assigned to major types of adult PNs using the same method.

*Selection of key parameters for E16.5 and P5 data analysis.* At E16.5, different PN types are establishing connections with their respective central and peripheral targets, so we reasoned that major PN types should be distinguishable at tran-scriptomic level at this stage, as recently suggested in a study by the lab of Joriene de Nooij[41]. For the initial clustering of E16.5 data, a low resolution of 0.3 was chosen and identified 3 major clusters (Fig. 5a). Since heterogeneity was observed within the major clusters, we increased the resolution to 0.8, resulting in seven clusters, which appeared to be in agreement with the structure of the tSNE plot (Supplementary Fig. 5b). For the analysis of P5 data, a resolution of 0.8 was used. Owing to small number of cells, cluster6 at P5 was selected manually based on its high specific $Chodl$ expression.

**Reporting summary**. Further information on research design is available in the Nature Research Reporting Summary linked to this article.

## Data availability
All data is available in the main text or the supplementary materials, apart from the scRNAseq transcriptomic data, which is accessible at GEO data repository (accession code: GSE156180). A browsable resource of E16.5, P5, and P54 PNs data is available at the lab website: https://ki.se/en/neuro/lallemend-laboratory.

Source data are provided with this paper.

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

## Acknowledgements

We thank Prof. Fan Wang and Dr. Joriene de Nooij for the *RGT* and *Egr3*<sup>WGA</sup> mouse lines, respectively, the CLICK imaging Facility supported by the Knut and Alice Wallenberg Foundation, the Uppsala Multidisciplinary Center for Advanced Computational Science (UPPMAX) for providing data storage resources, and Kuisong Song for technical implication in some results of Fig. 5j. We are also grateful to Prof. Sten Grillner for critical reading of the manuscript. This work was supported by grants from: Strat-Neuro, the Swedish Brain Foundation, Karolinska Institutet, the Swedish Research Council, KID funding (F.L. and S.H.); the Knut and Alice Wallenbergs Foundation (Wallenberg Academy Fellow), Ragnar Söderberg Foundation (Ragnar Söderberg Fellow in Medicine) and Ming Wai Lau Foundation (F.L.). F.L. is a Wallenberg Academy Fellow in Medicine and a MWLC investigator.

## Author contributions

F.L. conceived and supervised the project, with inputs from H.W. and S.H. H.W. and F.L. designed experiments; H.W., C.P., P.F., A.S., C.B., Y.W., K.K.Y.C., S.W., and P.J. performed experiments and analyzed data; R.M.Q., J.A.H., Y.X., K.M., J.R., C.B.G., and O.K. provided unique reagents and mouse models. H.W, S.H., and F.L. wrote the manuscript with inputs from all co-authors.

## Funding

## Competing interests

The authors declare no competing interests.
