## [Peer Review File · Nature Communications]

Reviewers' Comments:

Reviewer #1:

Remarks to the Author:

This well-written manuscript describes the use of deep single cell RNAseq to analyze the transcriptomes and classification of proprioceptive neurons (PNs) in the mouse DRGs, identifying 3 major molecular clusters that can be further subdivided into 8 subtypes. Through careful genetic labeling, retrograde tracing, and anatomical characterization, the authors correlated the 3 molecular groups to the 3 main functional types of proprioceptive afferents classically described in the field (Ia, Ib, and II). The study further characterizes the distribution of molecularly defined PNs across spinal segments, their developmental emergence in embryonic (E16.5) and neonatal (P5) mice, and their intriguing plasticity in adult mice after exercise training. Overall, the data for these experiments is of high quality and clearly presented, and the transcriptome resource has laid an important foundation for future research by enabling genetic access to the major PN subtypes. There are a few minor concerns with the manuscript:

1 – To isolate PNs, the authors use DRGs from PV-Cre;Ai14 mice, although PV-Cre has been shown to label subsets of LTMRs as well. This is even noted in the Methods section of the manuscript, as Ret+ and Ntrk2+ (as well as Sox10+ glia) clusters were identified and removed from subsequent analyses. Given the importance of the transcriptome data derived from PV-Cre-labeled PNs for the manuscript, this information should be clearly mentioned in the Results section.

2 – Figure 4 does not provide independent validation for the expression of the presented genes or their function as molecular mechanoreceptors in PNs, which makes it difficult to interpret in this analysis. For example, TrpC3 is shown to be enriched in some PN subtypes based on the authors' analysis, but other studies (e.g. Dong et al, 2017) have shown that TrpC3 is almost exclusively expressed in small diameter neurons and does not co-label with NF200. It is this reviewer's opinion that this figure be de-emphasized and moved to the supplement.

3 – Figure 6 proposes an interesting model (increased neuronal activity) for the plasticity of adult PNs, conversion from Ia1 to Ia2/3, but it does not provide enough evidence to support this speculative model or differentiate it from other possibilities (for example, increased expression and secretion of neurotrophic factors). A subtype identity switch from Ia1 to Ia2/3 PNs implies that the total number of Ia PNs should be constant. Therefore, one piece of evidence to support the proposed model would be providing raw counts of Ia PNs across sedentary and running mice, instead of only showing normalized fold changes. Without this piece of data, the author can't exclude a theoretical possibility that adult progenitors (very small number) give rise to a few Ia2/3 proprioceptors instead of conversion from Ia1 to Ia2/3. Ideally, evidence with subtype switching to Ia2/3 PNs from sparsely genetic labeling of Ia1 PNs and through direct manipulation of neuronal activity (chemogenetic activation or overexpression of excitatory channels) would be required to formally establish the model. This reviewer feels that the suitable mouse line may not be currently available and thus these types of experiments is beyond the scope of this manuscript. In this case, it would be helpful to include discussing the other possible mechanisms for this Ia PN subtype plasticity.

Reviewer #2:

Remarks to the Author:

The manuscript by Wu and colleagues from the laboratory of Francois Lallemand combines single cell RNAseq with several other techniques to characterize the phenotypes of proprioceptive neurons (PNs) in the adult DRG. Overall the study is very nicely done and provides exciting new data regarding proprioception. I have only a limited number of minor comments

Abstract: why is it unexpected to find 8 subgroups of PNs?

Line 76: I don't see any data about Doc2b in either figure 1 or extended figure 1. And then Chad is used as a marker for cluster 4 going forward.

Line 224: typo, remove "the" before development.

Line 247: was Vstm2b determined to be a marker of II-PNs based on process of elimination? This seems a bit suspect.

Line 302: if there is no overall increase in number of 1a-PNs but there is an increase in 1a2/3-PNs is the suggestion that the 1a1-PN subtype has decreased through conversion? Panel 6g suggests this, but is there any evidence for decrease in 1a-PVs?

Figure 1d, the tSNE clustering seems a bit at odds with the results in that cluster 1 seems more closely related to clusters 5 and 6 than with clusters 2 and 3. Similarly, cluster 7 seems to be as unique as cluster 4.

Figure 5i: the BRN3C labeling is not convincing. As a transcription factor, labeling should be, largely, confined to the nucleus, but the opposite appears to be the case in this image.

Reviewer #3:

Remarks to the Author:

This manuscript represents an important addition to our knowledge of propriospinal sensory neurons. In particular, it provides a new set of markers that could be used for selective genetic control over propriospinal sub-types, charts the development of these sub-types, and uncovers unexpected changes in the Ia sub-type after sustained exercise training. The depth of the analysis and strength of the data supporting claims is a little uneven, but with some additional analysis and re-writing, it could be an excellent, exciting, and frequently cited paper.

1. Identification of molecular "major groups": On what basis were the clusters split into the three major groups. From the tSNE in figure 1D, it is not clear that Cl.1 should be included in a "major group" with Cl.2 and Cl.3, or that Cl.7 should be included with Cl.5, Cl.6, and Cl.8. It is important that the groupings be well established for the rest of the paper. A dendrogram or overall correlation matrix to show the cluster relationships with each other should be shown to support that these groupings are not based on selected genes shown in Figure 1E. The (very helpful!) biological validation suggests that the authors have selected the correct major groups based on anatomy and connectivity, but this should be better justified (or questioned?) in the sequencing data.

2. Marker genes should be tested combinatorially to validate the patterns of gene expression for the major groupings and for sub-types (not simply the expression of these markers with general proprioceptor markers). This is particularly important for the "major group" markers as they are used for subsequent analysis. For example, Chad and Pou4f3 are both present in Cl.4 and Cl.8 (but are only considered as markers for Cl.4 – it should be shown that they (mostly) do not overlap with Fxyd7. Fxyd7 (a marker of Cl.5-8) is shown to be expressed in a substantial fraction of Cl.3 and Cl.4 – it should be determined whether there are Fxyd7+/Lmcd1+ cells or Fxyd7+/Chad+. And Lmcd1 is present in a substantial fraction of Cl.4 and it should be tested whether it overlaps with Chad in tissue. (In addition, quantifications in Figure 3 based on Runx1 are difficult to interpret because RNA for Runx1 is quite broadly expressed.)

3. It would be helpful to include a table or summary figure for each of the 8 clusters showing the main marker genes used for subsequent work and other features such as soma size and connectivity, as in many cases the markers for each cluster change between experiments and figures. Broad markers (PV, Runx3, Whrn) and specific markers (Cpne4, Slc17a6) used for

downstream experiments should be added to Figure 1e or another figure, as well as Sox10, Ret, and Ntrk2.

4. Developmental analysis. This section needs to be strengthened through further analysis and/or description.

4.1 The clustering analysis for e16.5 and P5 is not well explained and it is not clear why the presented clusters were chosen. Were other resolutions tested? It appears that several e16.5 clusters and Cl.1 at P5 could be split further. Indeed, in Extended Data Fig. 5, these cell types are split further and the justifications that these are mainly developmental processes and not "cell type" differences is weak.

4.2 The putative "identification" of Ia and Ib subtypes for the e16.5 clusters is not strong. The use of "local" RhDex injected into the ventral horn to label Ia neurons is supportive, but not a very strong way to define these cells, as it is very difficult to restrict tracers to only the MN area (indeed, the methods mention the ventral horn). It seems that a very mixed group of PN subtypes was labeled and the authors rely on over-representation of one marker for this claim. Similarly, Doc2b-RFP was detected in nearly a third of Ia/II PN neurons (and again, were defined by Pou4f3 which is also in Cl.8 in adults) and the expression at lower cervical vs upper cervical segments is not enough to make a general statement about limb levels (vs thoracic or sacral). The discussion around this section should mention and consider these caveats.

4.3 The data for the claim of "late" Ia maturation vs "early" II maturation are a little confusing. Perhaps more cells would enhance the resolution of Cl.1? As it is Calb1 and Runx1 do seem to be expressed at the right and left sides of the cluster. Relatedly, the differences in Extended Data Fig. 6c is confusing and an overall correlation matrix would be helpful. Also, the authors emphasize the Ib/II genes that decrease in "Ia" neurons and this is clear, but it seems that II-PNs have similar absolute levels of markers for other subtypes.

Response to reviewers' comments

We are very glad that the reviewers find our study of high quality and very important for the field and are grateful for their constructive comments which we think have helped us to consolidate the study. We have performed additional experiments, analysis and revised the paper according to their remarks. Please find below the detailed answers to each point raised.

Reviewer #1 (Remarks to the Author):

Overall, the data for these experiments is of high quality and clearly presented, and the transcriptome resource has laid an important foundation for future research by enabling genetic access to the major PN subtypes. There are a few minor concerns with the manuscript:

1 – To isolate PNs, the authors use DRGs from PV-Cre;Ai14 mice, although PV-Cre has been shown to label subsets of LTMRs as well. This is even noted in the Methods section of the manuscript, as Ret⁺ and Ntrk2⁺ (as well as Sox10⁺ glia) clusters were identified and removed from subsequent analyses. Given the importance of the transcriptome data derived from PV-Cre-labelled PNs for the manuscript, this information should be clearly mentioned in the Results section.

A description is now included in the Result section, under the title “Deep single cell RNAseq reveals diversity of adult PNs”: “*In total 1109 PNs (Whrn⁺, Pvalb⁺, Runx3⁺, Ntrk3⁺ and Etv1⁺) passed quality control with an exceptionally high gene coverage (~11,000 genes detected per cell) (Extended Data Fig. 1b,c), allowing deep analysis of their molecular profiles. A small number of mechanoreceptors (Ntrk2⁺ or Ret⁺, Pvalb^{low} and Whrn⁻) were also identified and thus excluded from the subsequent analysis (Extended Data Fig. 1d).*”

2 – Figure 4 does not provide independent validation for the expression of the presented genes or their function as molecular mechanoreceptors in PNs, which makes it difficult to interpret in this analysis. For example, TrpC3 is shown to be enriched in some PN subtypes based on the authors' analysis, but other studies (e.g. Dong et al, 2017) have shown that TrpC3 is almost exclusively expressed in small diameter neurons and does not co-label with NF200. It is this reviewer's opinion that this figure be de-emphasized and moved to the supplement.

Single cell RNAseq has been frequently used as a reliable readout of gene expression in addition to the classic immunohistochemistry or *in situ* hybridization, especially with highly sensitive Smartseq techniques. It has frequently served as an entry for unveiling the molecular mechanisms of cell functions (Paul et al., 2017; Zheng et al., 2019), or has been used (for instance in patchseq) to confirm earlier predictions of gene expression based on classic immunostaining or channel neurophysiology (Fuzik et al., 2016). Thus, many studies from the last few years have shown that to a large extent, gene expression data from scRNAseq data was showing very few, if any, false positive transcriptional features. Moreover, in Fig. 4, we have used a stringent cut-off for a gene expression to be considered positive and to be shown or discussed in the text, considering only the 50% most expressed genes from our dataset to be represented and discussed (see Methods for details, under the title “Differential expression analysis”). We have now added a sentence in the Result section, under the title “Molecular signatures of neuronal communication ...” to express the need for further confirmation of their expression *in situ*: “*We further categorized those genes according to their functions and, while further analysis will be necessary to confirm their expression in situ, this list of candidate genes could serve as a basis for understanding the different properties of PN subtypes*”. We also agree with the reviewer that the single cell RNAseq data alone is however insufficient for functional claims. We present Figure 4 solely as a resource to facilitate the

interpretation/usage of this data set and enable the community to investigate further the functionalities of PNs. We do think that this resource is rather valuable for the community (as this was also the request from many neurophysiologists in the field) and hope that the reviewer could agree to keep its visibility as a main figure. This is usually presented as main figures in many recent scRNAseq studies (Paul et al., 2017; Haring et al, 2018), and helps the reader to correlate the gene expression to physiological features either known or to be further explored/confirmed.

For the example of the transcript *Trpc3*, it has been also shown in another single cell RNAseq study (Zeisel et al., 2018, <http://mousebrain.org/genesearch.html>) to be expressed at low level in PNs and at high level in non-peptidergic nociceptors. Whether it could be detected on tissue sections largely depends on the sensitivity of the methods, background and thresholds used. Anyhow, differences in levels of expression between the two populations might explain why *Trpc3* was not detected in NF200⁺ DRG neurons in Dong et al., 2017, which used classic FISH technique. Moreover, only 2 subtypes of PNs show relatively high expression in our data; these neurons represent about 20-25% of PNs (meaning 2 to 2.5% of all DRG neurons), which could easily be missed. To our own experience, RNAscope V2 has always succeeded in detecting the modestly expressed genes, but we regret that we could not test every gene in Figure 4 and hope the reviewer could understand.

3 – Figure 6 proposes an interesting model (increased neuronal activity) for the plasticity of adult PNs, conversion from Ia1 to Ia2/3, but it does not provide enough evidence to support this speculative model or differentiate it from other possibilities (for example, increased expression and secretion of neurotrophic factors). A subtype identity switch from Ia1 to Ia2/3 PNs implies that the total number of Ia PNs should be constant. Therefore, one piece of evidence to support the proposed model would be providing raw counts of Ia PNs across sedentary and running mice, instead of only showing normalized fold changes. Without this piece of data, the author can't exclude a theoretically possibility that adult progenitors (very small number) give rise to a few Ia2/3 proprioceptors instead of conversion from Ia1 to Ia2/3. Ideally, evidence with subtype switching to Ia2/3 PNs from sparsely genetic labeling of Ia1 PNs and through direct manipulation of neuronal activity (chemogenetic activation or overexpression of excitatory channels) would be required to formally establish the model. This reviewer feel that the suitable mouse line may not be currently available and thus these types of experiments is beyond the scope of this manuscript. In this case, it would be helpful to include discussing the other possible mechanisms for this Ia PN subtype plasticity.

In Figure 6c, we originally presented that the proportion of Ia-PNs among all PNs ($\frac{\# \text{Ia PNs}}{\# \text{all PNs}}$) in running mice did not differ from sedentary mice, which was strongly suggesting already that the number of Ia-PNs remain the same, and was arguing against any possible adult neurogenesis. We now strengthen this observation and conclusion by adding the raw count of Ia-PNs in the Results section, together with the observation of a decrease in the number of Ia₁-PNs (*Lipg*⁺) in the running mice DRG compared to the sedentary mice: “*The overall proportion of Ia-PNs (amongst all PNs) (Fig. 6c) and their total number (raw counts, normalized to sedentary condition: Sed 1 ± 0.13 and Run 0.98 ± 0.005) in brachial DRG of sedentary and running animals were identical. Importantly, in the running condition we observed a decrease in the proportion of Lipg⁺/Whrn⁺ PNs (labeling Ia₁-PNs) and an increase in CALBI⁺/PV⁺ PNs (labeling Ia_{2/3}-PNs) at limb level (Fig. 6c,d).*” Moreover, we have added a sentence about possible mechanisms of Ia-PN subtype plasticity in the Results section, last sentence: “*Whether this plasticity is controlled by activity-dependent programs or target-derived or systemic factors, e.g. cytokines or growth factors (from muscles or blood), is awaiting to be addressed.*”

Reviewer #2 (Remarks to the Author):

Overall the study is very nicely done and provides exciting new data regarding proprioception. I have only a limited number of minor comments:

1. Abstract: why is was it unexpected to find 8 subgroups of PNs?

Indeed, this is too subjective, is unnecessary and has now been removed.

2. Line 76: I don't see any data about Doc2b in either figure 1 or extended figure 1. And then Chad is used as a marker for cluster 4 going forward.

Thanks to the reviewer to notice this error, as it should be indeed "Chad" in Fig. 1 and associated extended data file instead of "Doc2b". This is now corrected.

3. Line 224: typo, remove "the" before development.

Done.

4. Line 247: was Vstm2b determined to be a marker of II-PNs based on process of elimination? This seems a bit suspect.

We agree with the reviewer that we should be more cautious when claiming Vstm2b to be II-PNs marker before this is validated more rigorously. We have rephrased that sentence and added a question mark for the II-PNs in Fig. 5k: "Together, these data suggest that *Tnfrsf8l3*⁺ and *Doc2b*⁺ E16.5 PNs identify Ia- and Ib-PNs, respectively. The remaining *Vstm2b*⁺ cluster thus likely represented II-PNs (Fig. 5k), however this will require validation in future studies using for instance genetic tracing."

5. Line 302: if there is no overall increase in number of Ia-PNs but there is an increase in Ia_{2/3}-PNs is the suggestion that the Ia₁-PN subtype has decreased through conversion? Panel 6c suggests this, but is there any evidence for decrease in Ia₁-PNs?

We thank the reviewer to bring this to our attention. We now have used *Lipg*⁺/*Whrn*⁺ by RNAscope to label Ia₁-PNs, which showed a significant decrease in the exercise condition. This result is added in Fig. 6c and Result section: "The overall proportion of Ia-PNs (amongst all PNs) (Fig. 6c) and their total number (raw counts, normalized to sedentary condition: Sed 1 ± 0.13 and Run 0.98 ± 0.005) in brachial DRG of sedentary and running animals were identical. Importantly, in the running condition we observed a decrease in the proportion of *Lipg*⁺/*Whrn*⁺ PNs (labeling Ia₁-PNs) and an increase in *CALB1*⁺/*PV*⁺ PNs (labeling Ia_{2/3}-PNs) at limb level (Fig. 6c,d)."

6. Figure 1d, the tSNE clustering seems a bit at odds with the results in that cluster 1 seems more closely related to clusters 5 and 6 than with clusters 2 and 3. Similarly, cluster 7 seems to be as unique as cluster 4.

Unfortunately this is inherent to t-SNE clustering which, while being a method of choice for producing a 2D visualisation of distinct clusters of data points, can fail to preserve the global geometry of the data. In fact, the relative position of clusters on the t-SNE plot is almost arbitrary and depends on random initialisation. Hence, while scRNAseq data sets often encompass several classes of cells, each further divided into subclusters/subtypes, as in our dataset, t-SNE plots can fail to capture such hierarchical structures and sometimes yield misleading visualisation. Having said that, the plot in Fig. 1d might not be misleading as one might think. Cl.2 and 3 for instance are indeed off-centred compared to Cl.1 but Cl.2 and 3 are found (fig. 3) to be of large soma size and most likely represent very dynamic Ia-PNs compared to Cl.1. Similarly, Cl.7 is also off-centred compared to Cl.5, 6 and 8, but once again the diameter of these neurons is larger than the other type II-PNs, indicating certainly a higher velocity. Hence, it is possible that some particular visible and still unknown biological characteristics of the subtypes (some linked to cell size and dynamic properties, others linked to target tissue being innervated such as limb muscles versus axial muscles) might be linked to gene expression differences that would have a large impact on the spatial organization of the clustering in the t-SNE plots.

7. Figure 5i: the BRN3C labelling is not convincing. As a transcription factor, labelling should be, largely, confined to the nucleus, but the opposite appears to be the case in this image.

We agree with the reviewer that the antibody of most (not all) transcription factors is expected to label the nucleus. However, to our own experience of DRG staining in adult, some antibodies of transcription factors, e.g. ISLET1, which stain only and specifically the nucleus of sensory neurons at developmental stages, in contrast show very high background staining in the membranes (or extracellular matrix) of postnatal DRG. In the case of BRN3C, although the background staining is high outside of the neurons or in their membrane (which is considered as strong non-specific background), the nuclear staining is very specific and enables the distinction between positive and negative staining. To make the nuclear staining of BRN3C more visible to the readers, we have changed it to grey scale in the Fig. 5i.

Reviewer #3 (Remarks to the Author):

This manuscript represents an important addition to our knowledge of propriospinal sensory neurons. In particular, it provides a new set of markers that could be used for selective genetic control over propriospinal sub-types, charts the development of these sub-types, and uncovers unexpected changes in the Ia sub-type after sustained exercise training. The depth of the

analysis and strength of the data supporting claims is a little uneven, but with some additional analysis and re-writing, it could be an excellent, exciting, and frequently cited paper.

1. Identification of molecular “major groups”: On what basis were the clusters split into the three major groups. From the tSNE in figure 1D, it is not clear that Cl.1 should be included in a “major group” with Cl.2 and Cl.3, or that Cl.7 should be included with Cl.5, Cl.6, and Cl.8. It is important that the groupings be well established for the rest of the paper. A dendrogram or overall correlation matrix to show the cluster relationships with each other should be shown to support that these groupings are not based on selected genes shown in Figure 1E. The (very helpful!) biological validation suggests that the authors have selected the correct major groups based on anatomy and connectivity, but this should be better justified (or questioned?) in the sequencing data.

We understand the concern of the reviewer, as t-SNE plots can fail to preserve the global geometry of the data. In fact, the relative position of clusters on the t-SNE plot is almost arbitrary and depends on random initialisation. Hence, while scRNAseq data sets often encompass several classes of cells, each further divided into subclusters/subtypes, as in our dataset, t-SNE plots can fail to capture such hierarchical structures and sometimes yield misleading visualisation. Having said that, the plot in Fig. 1d might not be misleading as one might think. Cl.2 and 3 for instance are indeed off-centred compared to Cl.1 but Cl.2 and 3 are found (fig. 3) to be of large soma size and most likely represent very dynamic Ia-PNs compared to Cl.1. Similarly, Cl.7 is also off-centred compared to Cl.5, 6 and 8, but once again the diameter of these neurons is larger than the other type II-PNs, indicating certainly a higher velocity. Hence, it is possible that some particular visible and still unknown biological characteristics of the subtypes (some linked to cell size and dynamic properties, others linked to target tissue being innervated such as limb muscles versus axial muscles) might be linked to gene expression differences that would have a large impact on the spatial organization of the clustering in the t-SNE plots. Anyhow, it remains that the grouping of the clusters into 3 major groups was not clearly explained, and we now have added a new sentence in the Methods section, under the title “Clustering and dimensional reduction”: “*These 8 clusters were then clustered into 3 major clusters based on similarity of expression of highly specific markers unbiasedly identified, and later confirmed by their expression in situ and through analysis of their innervation profile*”. We also used the most differentially expressed genes (which are not the most restricted genes, i.e. best markers) of each cluster (except for the Ib cluster) to represent and compare their expression amongst all clusters in a dotplot (see below). These results show that using another set of differentially expressed genes, we once again see a similarity of marker expression amongst Cl.1-3, which dissipate a bit as we move to specific genes expressed for instance in the Cl.2 then Cl.3, suggesting that Cl.1 might be a generic Ia population, from which differentiate the Cl.2 and then Cl.3. Similar observation is done for the Cl5-8, which reinforce the observations made in the Figure 1.

2. Marker genes should be tested combinatorially to validate the patterns of gene expression for the major groupings and for sub-types (not simply the expression of these markers with general proprioceptor markers). This is particularly important for the “major group” markers

as they are used for subsequent analysis. For example, Chad and Pou4f3 are both present in Cl.4 and Cl.8 (but are only considered as markers for Cl.4 – it should be shown that they (mostly) do not overlap with Fxyd7. Fxyd7 (a marker of Cl.5-8) is shown to be expressed in a substantial fraction of Cl.3 and Cl.4 – it should be determined whether there are Fxyd7+/Lmcd1+ cells or Fxyd7+/Chad+. And Lmcd1 is present in a substantial fraction of Cl.4 and it should be tested whether it overlaps with Chad in tissue. (In addition, quantifications in Figure 3 based on Runx1 are difficult to interpret because RNA for Runx1 is quite broadly expressed.)

We appreciate the suggestion of the reviewer and have now performed RNAScope of *Fxyd7/Lmcd1*, *Chad/Fxyd7*, *Chad/Lmcd1* and added the results with quantification in Extended Data Fig. 2a,b which is shown below. We have also added in the Result section: “Combinatorial expression analysis of the three markers showed minimal overlapping between *Fxyd7* and *Lmcd1* or *Fxyd7* and *Chad* (Extended Data Fig. 2a), and few *Chad*+ PNs expressing *Lmcd1* but systematically at very low levels (Extended Data Fig. 2a,b).”

For the reviewer’s question on RUNX1 as a marker for Ia₁-PNs in Fig. 3, we did notice indeed that its transcript is also expressed in other PN subtypes but at a much lower level compared to the Ia₁ population. For this reason, we used RUNX1 antibody, and not RNAScope *Runx1* probes, to target the Ia₁ population as the antibody was found inefficient in targeting very low *Runx1*-expressing cells, as it is the case for most antibodies we are using. Hence, RUNX1 was found restricted to a small population of PNs (about 25% at brachial level, Fig. 3d). Also, this population of PNs has a distinct soma size (compared to the other PNs with low *Runx1*) and spatial distribution along the spinal cord compared with any other

PN subtypes. We thus concluded that the RUNX1 antibody we used only or predominantly targeted the PNs with highest *Runx1* expression, i.e. Ia₁-PNs, whose proportion amongst PNs when using RUNX1 antibody is consistent with the quantification and proportion of the other PN populations. We have added an explanation in the Method section to notify this to the readers: “*Though Runx1 is statistically enriched in Ia₁-PNs, Runx1 transcript is also lowly expressed in other PN subtypes. However, we observed that RUNX1 as protein is detected in a specific population of PNs with distinct soma size and spatial distribution, which suggests that it labels predominantly Ia₁-PNs.*”

3. It would be helpful to include a table or summary figure for each of the 8 clusters showing the main marker genes used for subsequent work and other features such as soma size and connectivity, as in many cases the markers for each cluster change between experiments and figures. Broad markers (PV, Runx3, Whrn) and specific markers (Cpne4, Slc17a6) used for downstream experiments should be added to Figure 1e or another figure, as well as Sox10, Ret, and Ntrk2.

A table summarizing the different characteristics and markers of PN subtypes is now added as Figure 3l.

I

	Ia ₁	Ia ₂	Ia ₃	Ib	II ₁	II ₂	II ₃	II ₄
markers examined	RUNX1 Lipg	CALB1 Cpne4 Slc17a6 Hpse	CALB1 Cpne4 Slc17a6 Hpse Calb2	BRN3C Lipg	CART	Tox	Aldh1a3	Chodl
soma size	M	L	L	M	S	S	M	S
innervation	limb trunk	limb	limb (bag fiber)	limb trunk ↓	limb trunk	limb ↓ trunk	limb	limb

We have now added all the subtype markers used in the study to Fig. 1e (indicated in magenta). Glial markers (*Sox10*, *ErbB3*) are also added to this figure to show no glia contamination in the data set. The expression of pan-PN (*Whrn*, *Pvalb*, *Runx3*, *Ntrk3*, *Etv1*) and mechanoreceptor (*Ntrk2*, *Ret*) markers are presented in Extended Data Fig. 1c,d.

4. Developmental analysis. This section needs to be strengthened through further analysis and/or description.

4.1 The clustering analysis for e16.5 and P5 is not well explained and it is not clear why the presented clusters were chosen. Were other resolutions tested? It appears that several e16.5 clusters and Cl.1 at P5 could be split further. Indeed, in Extended Data Fig. 5, these cell types are split further and the justifications that these are mainly developmental processes and not “cell type” differences is weak.

We agree with the reviewer that the data analysis for E16.5 and P5 is not well explained in the Methods section, please note that the original code for data analysis at all stages will be released in our Github channel. We now have also included a brief description of the data analysis for E16.5 and P5 in the Methods section, especially about the justification of choosing key parameters: “*Selection of key parameters for E16.5 and P5 data analysis: At E16.5, different PN types are establishing connections with their respective central and peripheral targets, so we reasoned that major PN types should be distinguishable at transcriptomic level at this stage, as recently suggested in a study by the lab of Joriene de Nooij (J Neurosci, 2019). For the initial clustering of E16.5 data, a low resolution of 0.3 was chosen and identified 3 major clusters (Fig. 5a). Since heterogeneity was observed within the major clusters, we increased the resolution to 0.8, resulting in 7 clusters which appeared to*

be in agreement with the structure of the tSNE plot (Extended Data Fig. 5b). For the analysis of P5 data, resolution=0.8 was also used. Due to small number of cells however, cluster6 at P5 was selected manually based on its highly specific *Chodl* expression.” We agree also with the reviewer that this heterogeneity observed at E16.5 should be better discussed in the main text. We added the following text in Result section: “While the 3 majors clusters could be split further into subclusters, those were only found at this early developmental stage and a gene ontology analysis of these subdivisions’ markers revealed a high enrichment in terms involved in synapse assembly, cell adhesion and axon guidance. This suggests that the up-regulation and differential expression of genes necessary for the assembly of sensorimotor circuit around E16.5 might generate transcriptional diversity resulting in transient subdivisions amongst the major cell types, as recently suggested for the developing visual system in *drosophila* (Ozel et al., 2020). Further developmental studies will however be necessary to explain the biological basis of the subcluster diversity of PNs at this stage”.

For the structural heterogeneity observed in cluster1 (Ia-PNs) at P5, we have also tried another method Pagoda for clustering to verify if this cluster could be split further (see below heatmap). And it does split into 3, however, these 3 sub-clusters were indistinguishable when comparing their top marker genes and were not consistent with the clustering observed in the tSNE plot, or with adult Ia subtypes, and therefore could not be assigned to obvious cell types.

4.2 The putative “identification” of Ia and Ib subtypes for the e16.5 clusters is not strong. The use of “local” RhDex injected into the ventral horn to label Ia neurons is supportive, but not a very strong way to define these cells, as it is very difficult to restrict tracers to only the MN area (indeed, the methods mention the ventral horn). It seems that a very mixed group of PN sub-types was labelled and the authors rely on over-representation of one marker for this claim. Similarly, *Doc2b*-RFP was detected in nearly a third of Ia/II PN neurons (and again, were defined by *Pou4f3* which is also in C1.8 in adults) and the expression at lower cervical vs upper cervical segments is not enough to make a general statement about limb levels (vs thoracic or sacral). The discussion around this section should mention and consider these caveats.

We agree with the reviewer about the technical limitations and the difficulties of targeting specific PN types at this developmental stage. For the retrograde tracing experiment with Rh-Dex, we have added a new text in the result section and Methods: “Note that ~90% of traced DRG neurons were PNs ($PV^+/RUNX3^+$), illustrating the limited spreading of the dextran”. And also another text in the Result section: “A small number of $Vstm2b^+$ and $Doc2b^+$ cells were also traced in our experiments (Fig. 5g). This could be explained by the diffusion properties of the dextran which might be taken up by few nerve endings of other PNs terminating in the vicinity of the ventral horn of the spinal cord at this early stage (see Methods for details) but also by the incomplete cell-type specificity of the markers used which

at this stage showed enrichment but not unique expression in a given cluster (Fig. 5b).” We would like to thank the reviewer to have brought up this to our attention, as we realized that *Vstm2b* population of PNs could be easily segregated into a *Vstm2b*^{HIGH} and a *Vstm2b*^{LOW}, which were found to distinguish the Cl.1 from the Cl.2, respectively (see the picture below). This is explained in the Methods section. And we have redone the quantification of *Vstm2b* in Fig. 5g considering only the *Vstm2b*^{HIGH} PNs.

For the genetic tracing of Ib-PNs using *Doc2b*^{ddCre};*Ai14*, we have added in the Result section: “Notably, we observed that almost 90% of Ib-PNs (*BRN3C*⁺) were labeled by RFP at brachial level (Fig. 5i), while less than 30% of Ia- and II-PNs were positive, strongly suggesting that *Doc2b* marked mostly the Ib-PNs at E16.5. The small fraction of traced Ia- or II- PNs in *Doc2b*^{ddCre};*Ai14* mouse might be the result of a baseline recombination often seen in *ddCre* mouse lines (see Methods for details). Additionally, at E16.5, *Doc2b*⁺ PNs were found specifically enriched in DRG innervating the limbs (Fig. 5j), another characteristic of Ib-PNs (Fig. 3b). Together, these data suggest that Cl.2 and Cl.3 of E16.5 PNs identify Ia- and Ib-PNs, respectively, and the remaining Cl.1 cluster thus most likely represented II-PNs (Fig. 5k).” For the spatial distribution of *Doc2b*⁺ PNs, we have analysed also the thoracic (T2-T12) and lumbar (L1-L5) regions, and data are now added to the Fig.5 (+ see below panel j) and in the main text: “Additionally, at E16.5, *Doc2b*⁺ PNs were found specifically enriched in DRG innervating the limbs (Fig. 5j), another characteristic of Ib-PNs (Fig. 3b).”

4.3 The data for the claim of “late” Ia maturation vs “early” II maturation are a little confusing. Perhaps more cells would enhance the resolution of Cl.1? As it is Calb1 and Runx1 do seem to be expressed at the right and left sides of the cluster. Relatedly, the differences in Extended Data Fig. 6c is confusing and an overall correlation matrix would be helpful. Also, the authors emphasize the Ib/II genes that decrease in “Ia” neurons and this is clear, but it seems that II-PNs have similar absolute levels of markers for other subtypes.

We understand the concern of the reviewer, however 479 PNs have been sequenced, among which a total of 179 Ia-PNs, which together with the high sequencing depth, provide a high statistical power to detect cellular heterogeneity, even at the subtype level. When analysing the Runx1 positive cells within the cl.1, their distribution is actually spread over the whole cluster, without any distinction of subclustering, with as many positive cells on either side of the cluster. For the reviewer’s question about Ib/II genes decrease in Ia-PNs but Ia/Ib genes remain the same level in II-PNs from P5 to P54, we would like to conclude from this analysis that Ia-PNs mature after P5 through the repression of Ib/II genes, while this process is done in II-PN–s before P5. Regarding the Extended Data Fig. 6c, it depicts square root scale for the Y axis (now added in the figure legend). In fact, in P54 II-PNs, the values of the median of the expression of the 50 marker genes is 3.8 for Ia markers, 0.6 for Ib markers and 6.3 for II markers, showing a significant difference between the type II versus non-type II markers.

Reviewers' Comments:

Reviewer #1:

Remarks to the Author:

The author has addressed all major concerns. The manuscript is ready for publication.

Reviewer #2:

Remarks to the Author:

The authors have addressed all of my concerns. I support publication of this manuscript.

Reviewer #3:

Remarks to the Author:

The authors have addressed all of my concerns in a satisfactory manner.